# Hotspot propensity across mutational processes

Claudia Arnedo-Pac [1,2], Ferran Muiños [1,2], Abel Gonzalez-Perez [1,2] & Nuria Lopez-Bigas [1,2,3,4]

## Abstract

The sparsity of mutations observed across tumours hinders our ability to study mutation rate variability at nucleotide resolution. To circumvent this, here we investigated the propensity of mutational processes to form mutational hotspots as a readout of their mutation rate variability at single base resolution. Mutational signatures 1 and 17 have the highest hotspot propensity (5–78 times higher than other processes). After accounting for trinucleotide mutational probabilities, sequence composition and mutational heterogeneity at 10 Kbp, most (94–95%) signature 17 hotspots remain unexplained, suggesting a significant role of local genomic features. For signature 1, the inclusion of genome-wide distribution of methylated CpG sites into models can explain most (80–100%) of the hotspot propensity. There is an increased hotspot propensity of signature 1 in normal tissues and de novo germline mutations. We demonstrate that hotspot propensity is a useful readout to assess the accuracy of mutation rate models at nucleotide resolution. This new approach and the findings derived from it open up new avenues for a range of somatic and germline studies investigating and modelling mutagenesis.

Keywords Mutational Hotspots; Mutational Hotspot Propensity; Mutation Rate Variability; Mutation Rate Variability at Single-nucleotide Resolution; Mutational Signatures
Subject Categories Cancer; Genetics, Gene Therapy & Genetic Disease

## Introduction

Mutations are heterogeneously distributed along genomes, following the interplay between DNA damage and DNA repair. Their specific interaction defines what we know as mutational processes, with heterogeneous activity along the nucleotide sequence. Understanding these mechanisms underlying somatic and germline mutagenesis, in humans and other species, has profound implications for the biological and biomedical sciences.

At the large (e.g., megabase) scale, the variability of mutation rates of different mutational processes is known to be associated with features such as DNA accessibility, transcriptional activity and replication timing (Polak et al, 2015; Stamatoyannopoulos et al, 2009; Schuster-Böckler and Lehner, 2012; Lawrence et al, 2013). Such covariates have been successfully used to model neutral mutagenesis at this 'low' resolution (Lawrence et al, 2013). At a more local scale, the mutation rate depends upon nucleotide sequence composition (Nik-Zainal et al, 2012; Alexandrov et al, 2013a, 2020), which is also included in such models (Lawrence et al, 2013; Martincorena et al, 2017), as well as upon local—up to a few Kbp—scale chromatin features including nucleosome occupancy (Pich et al, 2018), transcription factor binding (Sabarinathan et al, 2016; Perera et al, 2016; Kaiser et al, 2016; Katainen et al, 2015; Guo et al, 2018), and non-canonical DNA secondary structures (Zou et al, 2017; Georgakopoulos-Soares et al, 2018). A few of these local chromatin features are known to interact with particular mutational processes to increase mutation rates within specific nucleotides. This is the case of UV-light damage at the binding sites of ETS-family of transcription factors (Fredriksson et al, 2017; Elliott et al, 2018; Mao et al, 2018) and APOBEC3A mutations at ssDNA loops within DNA hairpins (Hess et al, 2019; Buisson et al, 2019; Shi et al, 2020). Despite the knowledge accumulated to date on the genomic features that influence the mutation rate, the ultimate goal of determining how much and why the rate of mutation contributed by different processes vary at single base resolution (i.e., at each unique position in the genome) remains a major challenge (Hess et al, 2019).

The current lack of accurate estimates of the mutation rate at base resolution is primarily caused by the sparsity of observed mutations across the 3 billion base pairs (bp) of the human genome, despite pooling the thousands of whole genomes currently available in the public domain. For the same reason, we have not yet been able to comprehensively characterise the determinants of the variability of the mutation rate at this scale (Gonzalez-Perez et al, 2019; Supek and Lehner, 2019). We are thus in need of novel ways to study mutagenesis at single base resolution.

Here, we propose to study mutation rate variability at single base resolution through the analysis of mutational hotspots, that is, recurrent mutations affecting the exact same genomic position across samples. This idea is based on the hypothesis that hotspots will reflect, apart from random sampling effects due to the finite size of the genome, genuine differences in the underlying mutation probability per unique genomic site for a given mutational process. We thus posit that tracking the rate of formation of hotspots will allow us to explore which mutational processes have a higher mutation rate variability at single base resolution. It will also allow us to quantify how much of this variability can be explained by large and small genomic features.

[1]Institute for Research in Biomedicine (IRB Barcelona), The Barcelona Institute of Science and Technology, Barcelona, Spain. [2]Centro de Investigación Biomédica en Red en Cáncer (CIBERONC), Instituto de Salud Carlos III, Madrid, Spain. [3]Institució Catalana de Recerca i Estudis Avançats (ICREA), Barcelona, Spain. [4]Department of Medicine and Life Sciences (MELIS), Universitat Pompeu Fabra (UPF), Barcelona, Spain. ✉E-mail: abel.gonzalez@irbbarcelona.org; nuria.lopez@irbbarcelona.org

In this study, we leveraged the whole-genome somatic mutations of more than 7500 tumours from 49 cancer types, and systematically detected and quantified the mutational processes creating passenger hotspots. We discovered that mutational processes active across tumours exhibit very different propensities to form hotspots, with signatures 1 (SBS1) and 17 (SBS17a and SBS17b) generating 5–78 times more hotspots than other common mutational processes when controlling for differences in their activities across tumours. We found that trinucleotide mutational probabilities, sequence composition and heterogeneity of mutation rates at 10 Kbp only explain a fraction of hotspot propensity among mutational signatures, ranging from 5% in SBS17 to 68% in SBS1. Conversely, by including genome-wide distribution of methylated CpG sites into SBS1 models, we get to explain 80–100% of SBS1 hotspot propensity. The high hotspot propensity of SBS1 is also observed in normal tissue samples from mammals, as well as across de novo germline mutations in the human population. Our work provides a new metric to study mutation rate variability at nucleotide resolution, highlighting the current difficulty to accurately estimate and model mutation rate variability across most mutational processes, with the exception of SBS1. These findings have strong implications for studies of basic mutagenesis as well as for those measuring positive or negative selection in cancer and evolution.

# Results

## Mutational hotspots across cancers

We developed a new method, named HotspotFinder, to identify and annotate unique genomic positions that are recurrently mutated (two or more times; i.e., hotspots) to the same alternate base (e.g., two C>T transitions) across the whole-genomes of 7507 sequenced primary and metastatic tumours from 49 cancer types (Fig. 1A–D; Dataset EV1 and EV2; Methods and Appendix Note 1). To avoid false positive hotspots due to sequencing, mapping or somatic mutation calling errors (Smith et al, 2016), careful filters of problematic genomic areas and positions containing population variants were applied (Methods and Appendix Note 1). Similarly, we excluded potential hotspots caused by positive selection by filtering out mutations overlapping the coding —and surrounding non-coding—sequence of known cancer driver genes (Sondka et al, 2018; Martínez-Jiménez et al, 2020) (Fig. 1E and Dataset EV3; Methods).

A total of 1,562,004 alternate-specific hotspots of four different types of mutations were identified across the set of specific cancer types (3,106,161 across the pan-cancer cohort): 1,361,631 corresponded to SNVs (87.2%), 125,657 to deletions (8.0%), 72,892 to insertions (4.7%), and 1824 to MNVs (0.1%) (Fig. 1F,G and Dataset EV4). Hotspots covered roughly 0.13% of the mappable hg38 reference genome (~2439 Mbp after excluding driver-associated regions; see Methods) and the vast majority (99.35%) were located in non-coding regions. In all cancer types except retinoblastomas at least one hotspot was observed. The majority of hotspots were small, comprising 2 or 3 mutated samples, with few exceptions (Fig. 1H; Appendix Figs. S1 and S2). Hotspots of insertions and deletions were particularly abundant (at similar or greater rates than SNVs hotspots) across cancer types with active indel

mutational processes (i.e., 19.3% and 30.1% insertion and deletion hotspot frequency in colorectal tumours, respectively) (Fig. 1F,G; Appendix Fig. S3). High rates of hotspots of insertions and deletions across other tumour types with few samples may be due to specific mutational processes such as mismatch repair deficiency, or sequencing and/or calling errors and biases across cohorts as shown in other studies (Priestley et al, 2019). Given that the vast majority of hotspots identified were composed of SNVs, and because errors in indels mapping and calling could lead to higher rates of false positive insertion and deletion hotspots (Priestley et al, 2019), we decided to focus on the study of SNV hotspots formation. Henceforth, we use hotspot as synonymous with SNV hotspot, unless otherwise specified.

While the number of hotspots per cancer type increased with sample size and mutation burden (Spearman's R = 0.93, $p = 9e-18$ and R = 0.72, $p = 2e-7$, respectively; Appendix Fig. S4A,B), similarly as predicted by theoretical models of homogeneous mutation rates across trinucleotides (Appendix Fig. S4C,D), hotspots appeared at different rates across tumour types (Appendix Figs. S5 and S6). Computing the number of SNVs required to generate one hotspot (hotspot conversion rates) showed large variability across cancer types, ranging from 21 mutations in melanomas to 3036 mutations in medulloblastomas (Appendix Fig. S7). This variability in conversion rates, together with the small fraction of hotspots shared between cancer types (Appendix Fig. S8), suggested that hotspots could capture differences in mutation rate heterogeneity at single base resolution across mutational processes active in distinct tissues.

## Propensity of mutational processes to form hotspots

To study the variability of the mutation rate at single nucleotide resolution for different mutational processes, we first analysed the enrichment of hotspots across different sequence contexts. We observed that, across cancer types, hotspots were differentially enriched for different types of nucleotide changes, and within particular trinucleotide sequences (Appendix Figs. S9 and S10). A similar observation was made for indel hotspots (Appendix Fig. S11).

We then estimated the number of hotspots generated by single base substitution (SBS) mutational signatures in each tumour type by conducting a de novo extraction and decomposition into the COSMIC (v3.2 GRCh38) signatures to facilitate comparisons across cancers (Fig. 2A; Appendix Fig. S12; Methods and Appendix Note 2, 3). SBS1 appeared as an important contributor of hotspots across all cancer types, particularly in tumours of the brain (69.6% of hotspots), pancreas (60.9%), prostate (42.3%) and colorectum (39.7%) (Fig. 2B; Appendix Fig. S13), in agreement with the enrichment of hotspots in C > T transitions in the NpCpG context (Appendix Figs. S9 and S10). SBS17a and b contributed a large proportion of hotspots in oesophageal cancers (16.8% and 72.3% of hotspots, respectively), stomach tumours (12.1% and 67.4%), and, to a lesser degree, colorectal cancers (1.51% and 18.6%) (Fig. 2B; Appendix Fig. S13), in accordance with hotspot enrichment for T > G transversions and T > C transitions, especially within CpTpT contexts in gastrointestinal tumours (Appendix Figs. S9 and S10). SBS5, an age-related signature of unknown aetiology, ubiquitous across cancer types, SBS2 and SBS13 (APOBEC-related), SBS4 (related to tobacco smoking), and SBS7a (caused by UV light

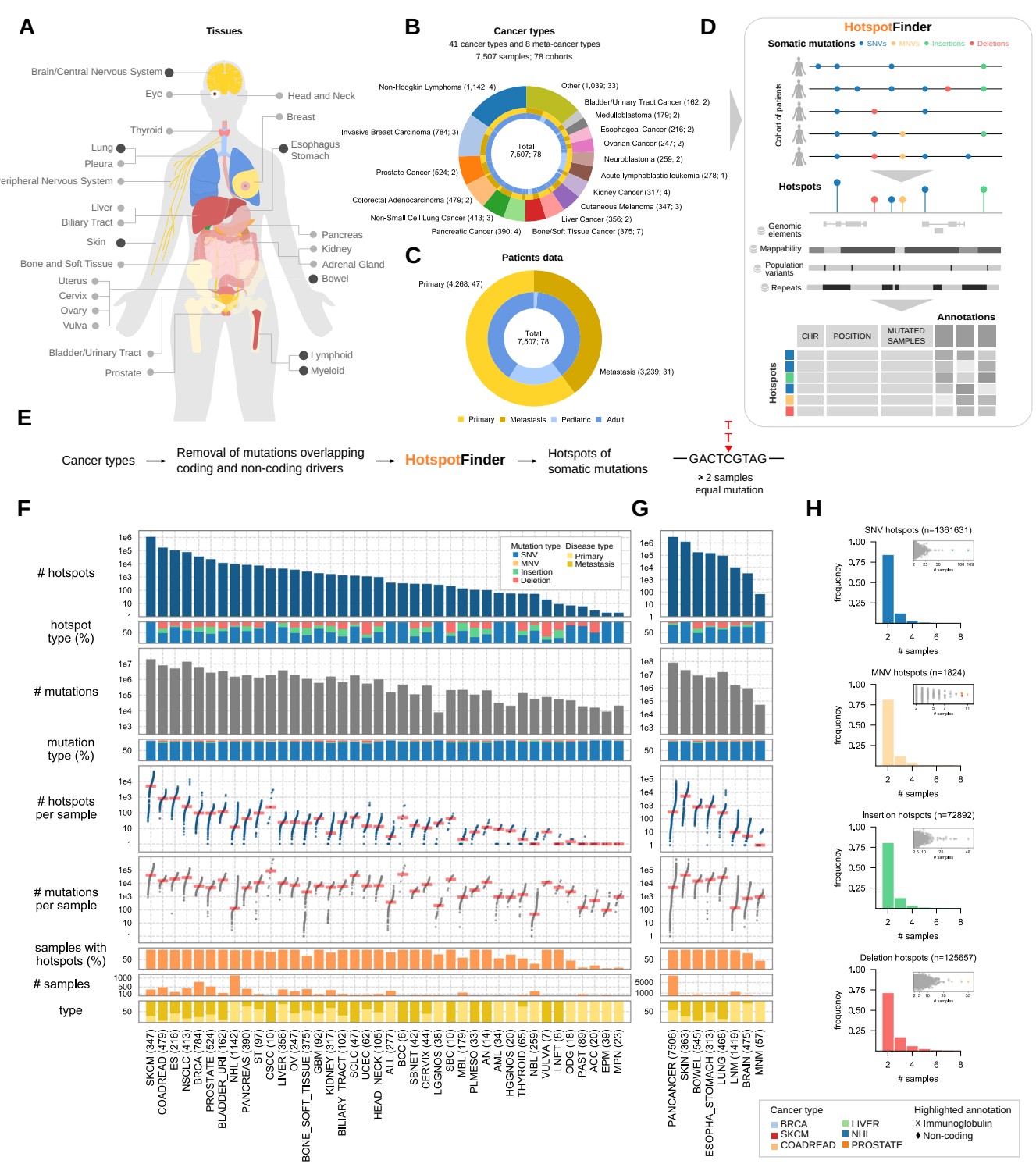

damage) were responsible for a high proportion of hotspots of specific cancer types (Fig. 2B; Appendix Fig. S13). The analysis of the extended context (up to 21 bp) of hotspots per signature showed that despite small preferences for specific nucleotides outside the trinucleotide context (e.g., SBS7a and b (Alexandrov et al, 2020), SBS17a and b (Secrier et al, 2016; Christensen et al,

2019), and SBS93), the contribution of the trinucleotide sequence clearly dominates all of them, suggesting it plays the preponderant role in the formation of hotspots as is the case with mutations (Alexandrov et al, 2020) (Appendix Fig. S14).

The burden of hotspots contributed by each mutational signature showed, as expected, a good correlation with its activity

◀ **Figure 1.  Identification of hotspots across cancers.**

(**A**) Cancer types analysed depicting specific cancer types and meta-cancer types in light and dark grey, respectively. Meta-cancer types include at least two specific cancer types (see Methods and Dataset EV1 and EV2 for further details). (**B**) Number of patients and sequencing cohorts among specific cancer types. (**C**) Detail of the number of primary, metastatic, adult and paediatric tumours analysed. (**D**) Schematic overview of HotspotFinder, a new algorithm to identify hotspots of somatic mutations (Methods; Appendix Note 1). (**E**) Overview of the steps for hotspots identification. (**F, G**) Summary of total hotspots identified across specific cancer types (**F**) and meta-cancer types (**G**) with at least one hotspot. Number of samples per cancer type after filtering are shown in parentheses. Note that 1 sample belonging to ALL, LNM and PANCANCER was excluded from the analysis since no mutations were left in the mappable genome after filtering (Methods). (**H**) Histograms of hotspot size (number of mutated samples per hotspot) considering only hotspots from specific cancer types. Embedded dotplots show hotspot sizes per individual hotspot, where the shape and the colour represent the overlapping genomic element and the cancer type where the hotspot was identified, respectively. Cancer types are listed as follows: Acute lymphoblastic leukaemia (ALL), acute myeloid leukaemia (AML), adrenocortical carcinoma (ACC), anal cancer (AN), basal cell carcinoma (BCC), biliary tract (BILIARY_TRACT), bladder/urinary tract (BLADDER_URI), bone/soft tissue (BONE_SOFT_TISSUE), bowel (BOWEL), CNS/brain (BRAIN), cervix (CERVIX), colorectal adenocarcinoma (COADREAD), cutaneous melanoma (SKCM), cutaneous squamous cell carcinoma (CSCC), endometrial carcinoma (UCEC), ependymoma (EPM), oesophageal cancer (ES), oesophagus-stomach cancers (ESOPHA_STOMACH), glioblastoma multiforme (GBM), head and neck (HEAD_NECK), high-grade glioma NOS (HGGNOS), invasive breast carcinoma (BRCA), kidney (KIDNEY), liver (LIVER), low-grade glioma NOS (LGGNOS), lung (LUNG), lung neuroendocrine tumour (LNET), lymphoid neoplasm (LNM), medulloblastoma (MBL), myeloid neoplasm (MNM), myeloproliferative neoplasms (MPN), neuroblastoma (NBL), non-Hodgkin lymphoma (NHL), non-small cell lung cancer (NSCLC), oligodendroglioma (ODG), ovarian cancer (OV), pan-cancer (PANCANCER), pancreas (PANCREAS), pilocytic astrocytoma (PAST), pleural mesothelioma (PLMESO), prostate (PROSTATE), retinoblastoma (RBL), skin (SKIN), small bowel cancer (SBC), small bowel neuroendocrine tumour (SBNET), small cell lung cancer (SCLC), stomach cancer (ST), thyroid (THYROID), vulva (VULVA).

and the proportion of samples at which it was found active (Appendix Fig. S15). In order to account for this in our analysis, we set out to implement a new metric, the propensity of a signature to form hotspots, which informs about its intrinsic inclination to generate hotspots independently of their overall cohort activity and sample-wise contribution to the mutation burden. The larger the hotspot propensity of a mutational process, the higher the variability of its mutation rates at single base resolution.

To test this new metric, we selected 14 mutational signatures with high activity in one or more cancer types (detailed criteria in Methods) and re-ran hotspot identification upon subsampling a fixed number of mutations contributed by each of them. Across a range of 10,000–30,000 mutations sampled from 100 tumours (at equal number of mutations per sample), we observed approximately 1 to 2 orders of magnitude more hotspots contributed by SBS17b, SBS17a and SBS1 than by the other eleven mutational signatures studied across tumour types (Fig. 2C–F). Specifically, for 30,000 mutations across 100 samples, a median of 78, 72 and 39 hotspots were observed for SBS17b, SBS17a and SBS1, respectively (Fig. 2F). Conversely, SBS7a, SBS18, SBS8, SBS93, SBS2, SBS13, and SBS7b contributed 3–7 hotspots, and SBS5, SBS3, SBS40 and SBS4 generated 1–2 hotspots (Fig. 2F). That is, SBS17b, SBS17a and SBS1 contributed 5–78 times more hotspots than the other signatures under the same conditions. Equivalent results were observed with larger sampling (Appendix Fig. S16). In an orthogonal—subsampling independent—calculation of the propensity to form hotspots employing the fold change of mutations contributed by each signature across tumours inside and outside hotspots, we obtained very similar results (Fig. EV1; Appendix Fig. S17; Appendix Note 4). In summary, SBS17b, SBS17a and SBS1 show the highest propensity to form hotspots among mutational processes commonly active in human tissues.

## Signature profile unevenness and trinucleotide abundance affect hotspot formation

We hypothesised that one possible reason for the disparity in the propensity to form hotspots of different mutational processes could be the number of genomic positions available to each of them. This is determined, in the first place, by the particular trinucleotide mutational probabilities of the signature, which give rise to a particular shape (i.e., skewness or unevenness vs uniformity or evenness) of its trinucleotide profile. We measured the unevenness of the activity of signatures across trinucleotides as the entropy of their mutational profile normalised by the abundance of each trinucleotide in the human genome, which informs about the mutational probability of the signature across trinucleotides irrespective of the genome composition (Dataset EV5). The more uneven the profile of a signature, the lower its entropy (Figs. 3A and EV2A). The three signatures with the highest propensity to form hotspots across tissues (SBS1, SBS17a and SBS17b) possess low entropy profiles (Fig. 3A,B; Methods). Conversely, signatures with even profiles, such as SBS3 and SBS5, did not display a high propensity to form hotspots (Fig. 3A,B). These results suggest that the fewer active trinucleotides in a mutational signature (the more uneven its profile), the more likely it is that two mutations contributed by the process map to the same genomic position.

The availability of the 96 trinucleotides in the human genome also influences the propensity of different mutational signatures to form hotspots. To investigate the combined effect of genomic trinucleotide abundance and signatures prolife unevenness, we determined the expected (based solely on the trinucleotide substitution probabilities) number of hotspots formed by mutations contributed by several processes (see details in Methods and Appendix Note 5). Following this assumption, 5.3–126 times more hotspots were expected to be contributed by SBS1 than by any other signature (Figs. 3C and EV2B). The explanation for this is that the four NpCpG trinucleotides targeted by SBS1 are comparatively depleted in the human genome (0.4–0.5% NpCpG vs 1.9–7.9% non-NpCpG in mappable bins; Figs. 3D and EV2C,D). In contrast, the NpTpT and CpTpN trinucleotides, which respectively concentrate the activity of SBS17b and SBS17a, show average (or above average) representation in the human genome (Figs. 3D and EV2C,D). Interestingly, for all mutational signatures analysed, we observed more hotspots than expected, with the highest differences for SBS17b and SBS17a (Fig. 3E;

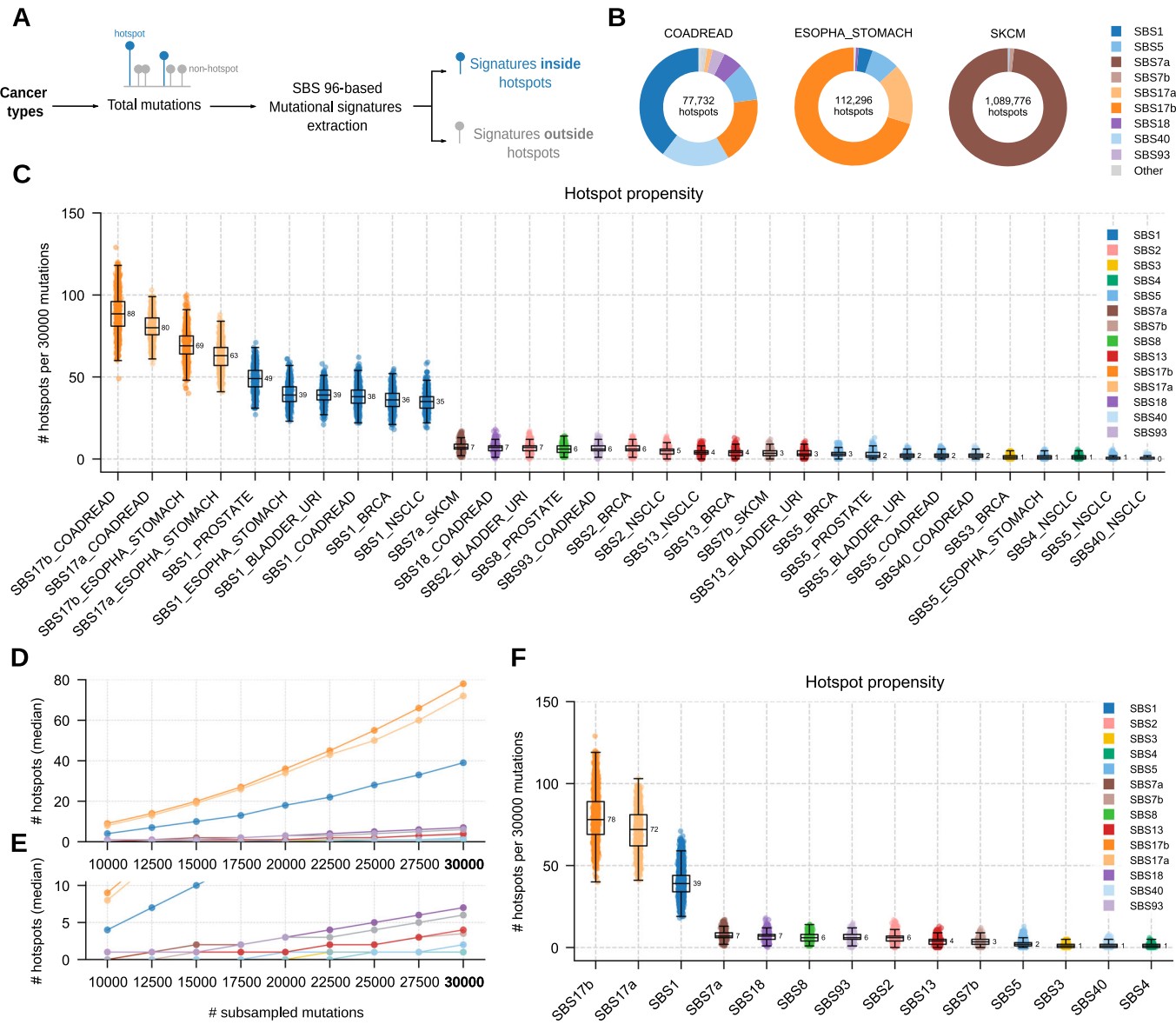

**Figure 2. Mutational processes with increased propensity to form hotspots.**

(A) Graphical definition of the analysis of mutational signatures in the sets of mutations inside and outside hotspots. (B) Pie charts depicting the relative number of hotspots observed per signature in the cancer type. (C) Number of hotspots per signature and cancer type observed by subsampling 30,000 total mutations (300 mutations/sample, 100 samples) within each group in the set of mappable megabases. Boxplot centre depicts the median, and the lower and upper bounds of the box represent the 1st and the 3rd quartiles, respectively. Whiskers show 1.5 times the interquartile range (IQR) below and above the 1st and 3rd quartiles, respectively. The underlying dots comprise the entire range of the distributions represented by the boxplots, including its minimum and maximum. Each dot represents a subsample in the cancer type where a signature's hotspot propensity was analysed (Methods). Signatures are sorted in descending order according to their mean number of observed hotspots. (D) Median number of observed hotspots per signature across subsamples at different mutation burden (100–300 mutations/sample, 100 samples). In order to obtain signature-level estimates, subsamples across different cancer types were merged as listed in Methods. (E) Zoom into 0–10 median hotspots from (D). (F) Number of hotspots per signature observed within 30,000 subsampled mutations (300 mutations/sample, 100 samples) across cancer types. Boxplot centre depicts the median, and the lower and upper bounds of the box represent the 1st and the 3rd quartiles, respectively. Whiskers show 1.5 times the IQR below and above 1st and 3rd quartiles, respectively. The underlying dots comprise the entire range of the distributions represented by the boxplots, including its minimum and maximum. Each dot represents a subsample within one of the cancer types where a signature's hotspot propensity was analysed, merging data shown in (C) (Methods). Signatures are sorted in descending order according to the mean number of observed hotspots.

Methods). Since the number of hotspots computed via the theoretical model only accounts for the unevenness of the mutational profile of the signature and the abundance of trinucleotides in the genome, we sought to quantify the influence of other variables to hotspot propensities.

## Hotspot propensity is differentially impacted by large-scale chromatin features across signatures

We reasoned that another factor underlying the differences in hotspot propensity across signatures could be the uneven distribution of their

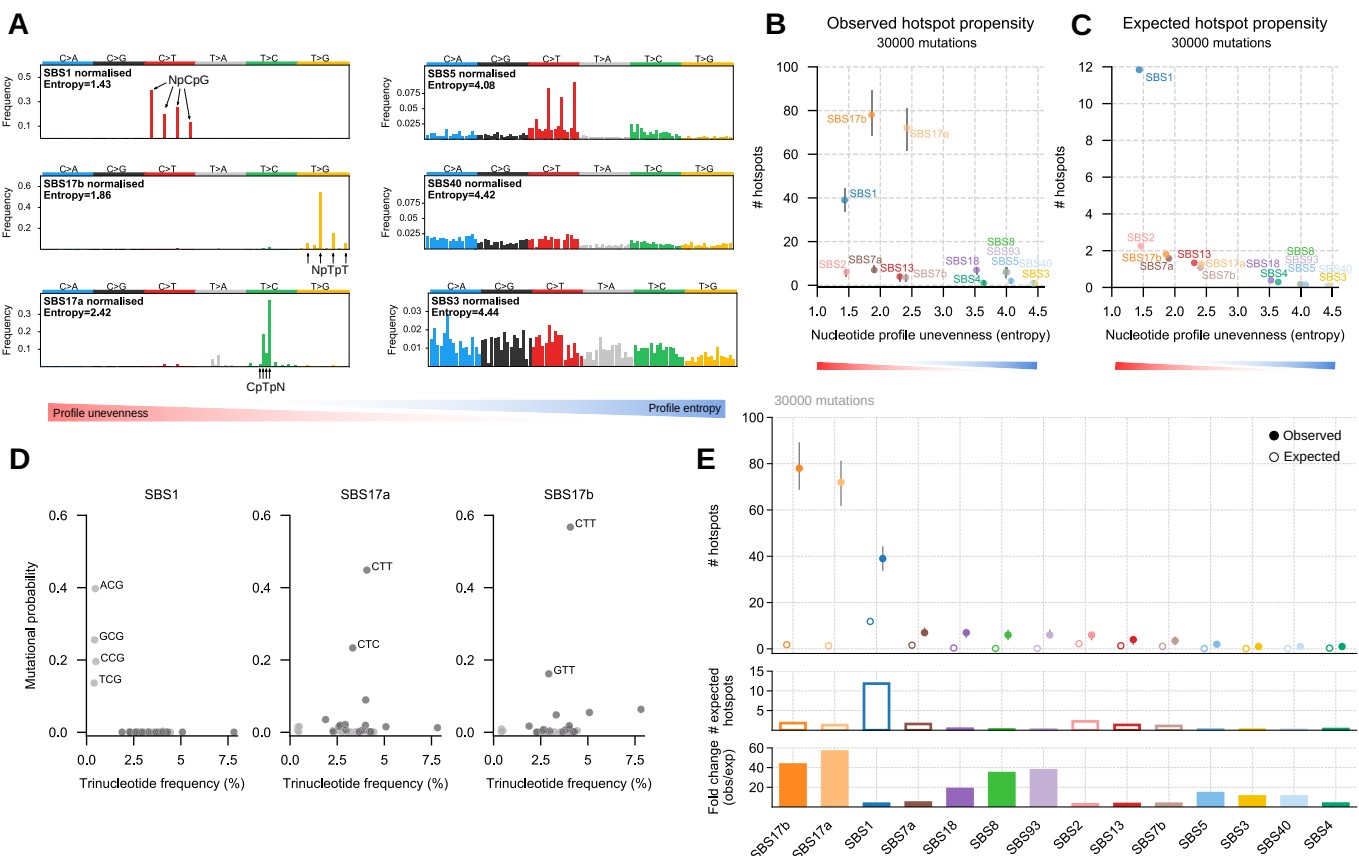

**Figure 3. Contribution of trinucleotide mutational probabilities and trinucleotide abundance to hotspots formation across signatures.**

(A) Normalised trinucleotide profiles of SBS1, SBS17a, SBS17b, SBS40, SBS5 and SBS3; additional signatures are shown in Fig. EV2. (B) Number of observed hotspots versus the entropy of the normalised signature profile. Observed hotspot propensity was computed by subsampling 30,000 total mutations (300 mutations/sample, 100 samples) within mappable megabases. Error bars show the IQR of the number of hotspots observed across subsamples. (C) Number of expected hotspot versus the entropy of the normalised signature profile. Expected data was generated using the model of homogeneous distribution of trinucleotide-specific mutation rates across the genome for 300 mutations/sample and 100 samples within mappable megabases, therefore it is comparable to the data in (B). (D) Mutational probability of trinucleotides per signature versus their frequency within mappable megabases. The mutational probability for each trinucleotide was obtained by merging those from the respective three alternates given by the normalised signature profile. (E) Number of observed and expected hotspot propensity per signature (top and middle panels) and the fold change of observed to expected number of hotspots (bottom panel). Observed and expected hotspot propensities were calculated using 300 mutations/sample and 100 samples across mappable megabases. Dots show median hotspots; error bars show the IQR. Observed and expected data correspond to that shown in (B) and (C).

mutations along the human genome (Appendix Fig. S18). Several megabase scale chromatin features, such as replication time and chromatin compaction are known to play a role in this variability of the mutation rate along the genome (Polak et al, 2014; Stamatoyannopoulos et al, 2009; Schuster-Böckler and Lehner, 2012; Lawrence et al, 2013; Morganella et al, 2016; Tomkova et al, 2018; Otlu et al, 2023). To compute if the distribution of mutations at large scales influences the propensity of different signatures to form hotspots, we counted the number of mutations and hotspots contributed by each signature in autosomal mappable bins of length 1 Mbp. We found that the density of hotspots was positively correlated with that of mutations at the megabase scale (Appendix Fig. S19), as illustrated along chromosome 1 for SBS1 and SBS17b in colorectal cancers (Fig. 4A). As expected, hotspot density per megabase correlated with chromatin accessibility, replication time and level of transcription (Fig. 4A; Appendix Figs. S20, S21 and S22; Dataset EV6).

In order to quantify the potential effect of the large-scale genomic variability of a signature's activity in the formation of hotspots, we first computed the overdispersion of the distribution of its mutation rates across megabase bins (Fig. 4B; Methods). SBS1 mutations exhibited low overdispersion across megabase genomic bins. Others, like SBS17b, showed large variability in mutation counts at the megabase scale (Fig. 4B; Appendix Fig. S23). Actually, SBS1, SBS17a and SBS17b, the mutational processes with the highest propensity to form hotspots, appeared at opposite ends of the spectrum of megabase mutation overdispersion across all tested signatures (Fig. 4C). SBS17a and b exhibit the highest megabase mutation rate unevenness, followed by SBS18 and SBS4 (Fig. 4C). While differences in the interplay of mutational processes with chromatin features may underlie the dissimilar unevenness observed across signatures, it is worth noticing that the NpTpT trinucleotides targeted by SBS17a and SBS17b show a greater inter-megabase variability than the NpCpG targeted by SBS1 (Fig. EV2D).

Next, we compared the number of hotspots observed across 1 Mbp segments of the genome with that expected after accounting for the megabase distribution of mutations and the trinucleotide composition of each segment (Fig. 4D; Methods). While the expected number of hotspots across signatures increased compared

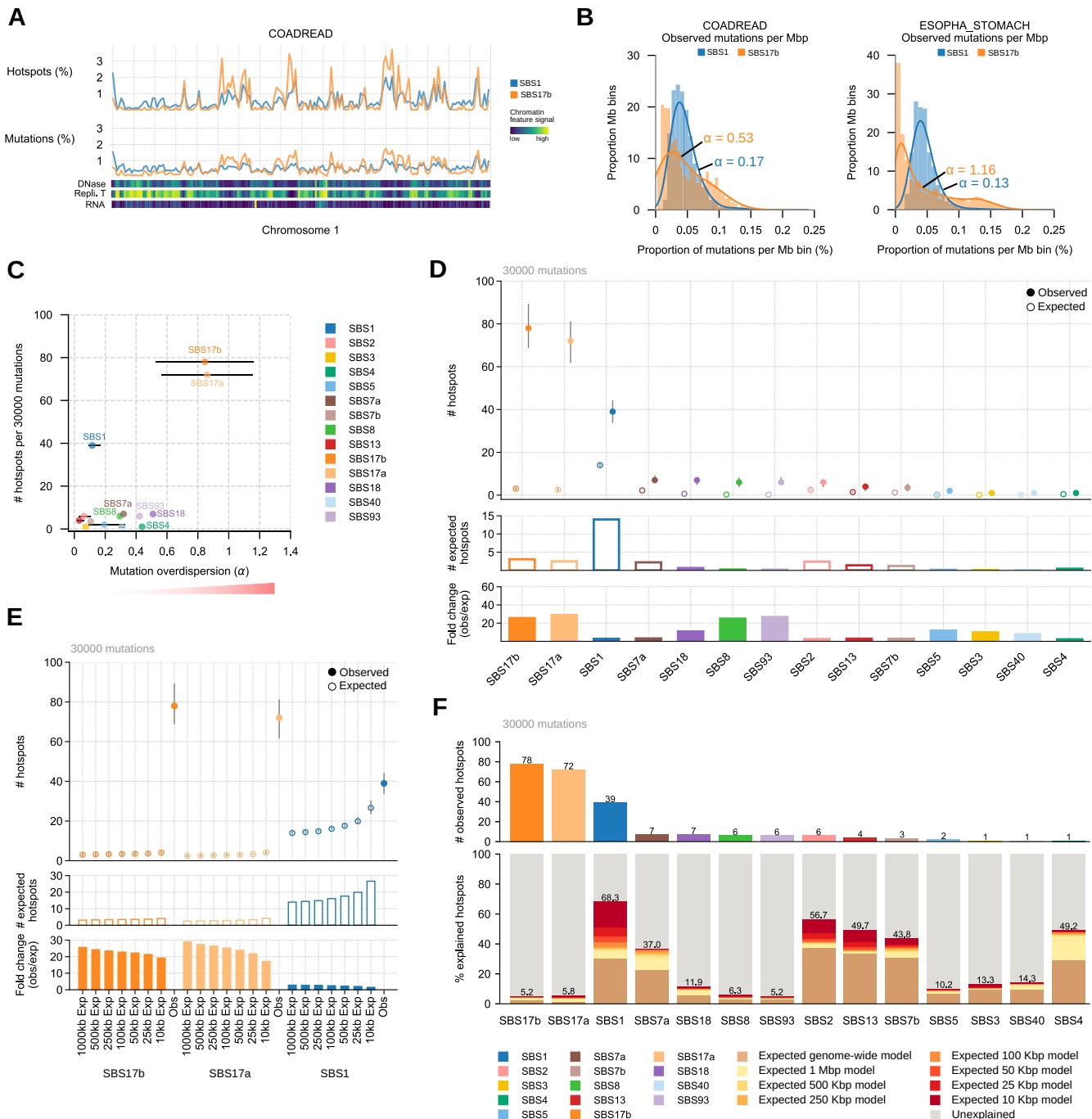

to that computed using only the signature mutational profile and the trinucleotide genome composition (Fig. 3E), the observed-to-expected hotspot fold-change was still greater than 1 for all signatures. In other words, the megabase-scale distribution of mutations contributed by different signatures has only a partial influence on their propensity to form hotspots.

We thus asked whether genomic features that affect the rate of mutations at scales smaller than 1 Mbp could explain part of the unaccounted hotspot propensity. To that end, we repeated the calculation of expected hotspots at sub-megabase bins of lengths 500, 250, 100, 50, 25 and 10 Kbp. Interestingly, we observed a monotonic increase of the expected number of hotspots across

signatures as the bin size decreased (Fig. 4E; Appendix Fig. S24), which is in line with the increase in the overdispersion of mutations (Dataset EV7). We next estimated the increasing contribution of large-scale covariates, on top of mutational probabilities of a signature and the trinucleotide composition of the genome, to the observed hotspot propensity. Nearly 70% of SBS1 hotspot propensity can be explained by the 10 Kbp models (Fig. 4F). The corresponding fraction is between 30 and 60% for SBS2, SBS13, SBS4, SBS7a and b. For other mutational processes, conversely, reducing the scale of the genomic bins at which the mutation rate variability is assessed does not result in appreciable increase in the fraction of hotspots explained by the models. Specifically, 94.2%

**Figure 4. Contribution of large-scale unevenness to hotspots formation across signatures.**

(A) Proportion of observed hotspots and mutations in colorectal cancers attributable to SBS1 and SBS17b across mappable megabases of chromosome 1. Normalised epigenomic signals of chromatin accessibility (DNase), replication timing (Repli. T) and expression (RNA) per megabase are shown below. (B) Distribution of the observed proportion of mutations across mappable megabases. Alpha values show the overdispersion of the negative binomial distribution fitted with mutation counts per megabase (Methods). (C) Number of observed hotspots versus the overdispersion (unevenness) of mutation counts across mappable megabases. Error bars show the range of mutation overdispersion of a signature across the cancer types where it has been analysed (Methods). (D) Comparison of observed versus expected number of hotspots per signature (top and middle panels) calculated with the megabase model accounting for trinucleotide composition and large-scale mutation rate variability. Observed and expected hotspot propensities were calculated using 300 mutations/sample and 100 samples across mappable megabases. Dots show the median number of hotspots across cancers. Error bars correspond to the IQR. The fold change of observed versus expected number of hotspots is shown below. (E) Comparison of observed and expected hotspot propensity for SBS1, SBS17a and SBS17b calculated across different bin sizes ranging from 1 Mbp to 10 Kbp long. Observed and expected hotspot propensities were calculated using 300 mutations/sample and 100 samples. Observed and expected dots show median number of hotspots; error bars show the IQR. Additional signatures are shown in Appendix Fig. S24. (F) Median observed hotspot propensity using 30,000 subsampled mutations (300 mutations/sample, 100 samples) (top). Fraction of the median observed hotspot propensity that is accounted for by the different expected hotspot propensity models calculated using 300 mutations/sample and 100 samples (bottom). The total percentage of hotspot propensity explained by the 10 Kbp-based model for each signature is shown above each bar.

and 94.8% of SBS17a and b hotspots, respectively, still remain unexplained, a proportion that is similar to those of SBS8 and SBS93. Extending the subsampling experiment to 60,000 mutations showed equivalent explainability gaps of hotspot propensities (Appendix Fig. S25). In summary, our results suggest that large-scale chromatin features, down to 10 Kbp, play differently important roles on hotspot formation across mutational processes. Although known mutation rate determinants can predict up to 70% of hotspot rates for some mutational processes, the knowledge gap is still noticeable across signatures (31.7–94.8%).

## Determinants of signature 17 hotspot propensity are largely unknown

Signatures SBS17a and b showed the greatest hotspot propensity, which remains unexplained, for the most part, by well-known determinants of mutation rate variability. We hypothesised that additional chromatin features at the local—below the 10 Kbp—scale (Gonzalez-Perez et al, 2019) may have an important influence on their hotspot propensity. We found that SBS17a and b hotspots appeared increased in colorectal (14.3 and 15.6 times) and oesophageal-stomach tumours (3.82 and 3.21 times) at CTCF binding sites with respect to their flanking sequences, significantly beyond the expectation from their sequence composition (Fig. 5A). These results are consistent with prior reports of CTCF binding sites bearing clusters of SBS17 mutations (Katainen et al, 2015; Guo et al, 2018; Otlu et al, 2023). Nevertheless, only 1.7% and 0.8% of SBS17a hotspots, and 1.3% and 0.5% of SBS17b hotspots in colorectal and oesophageal-stomach cancers, respectively, overlap CTCF binding sites (Fig. 5B). Yet, the majority of CTCF-overlapping hotspots in these two tumour types (86% and 53.5%) are attributed to the combined activity of these two signatures. Therefore, despite the contribution of CTCF binding sites to the expected hotspot rate, other still unidentified small-scale genomic features must also bear increased rates of SBS17a and b hotspots.

## Signature 1 hotspot propensity can be explained by tissue-matched methylation data

Similarly, we sought to identify additional factors to improve our estimates of SBS1 mutation rate variability at nucleotide resolution. Since the aetiology of SBS1 involves the spontaneous deamination of

5-methylcytosines (Alexandrov et al, 2013a), we reasoned that the differential methylation of CpG sites across the genome could be the reason behind the remaining unexplained hotspot propensity (Fig. 4F). Indeed, SBS1 hotspots, as well as mutations, observed across colorectal, oesophageal-stomach and non-small cell lung cancers are enriched for CpG sites that appear methylated in the respective tissues of origin (Fig. 5C–F). Motivated by this result, we next computed the expected hotspot propensity across these three tumour types adding the methylation status of CpGs to the features already considered in the models. We found that, as the size of genomic bins decreased, these methylation-aware models produced numbers of expected hotspots that approached the distribution of the observations for 30,000 randomly sampled mutations (Fig. 5G). Between 80% and 100% of the observed SBS1 hotspot propensity is ultimately explained by methylation-aware models at 10 Kbp bins (Fig. 5H).

## Signatures 1 and 17 high hotspot propensity in non-malignant somatic and germline tissues

SBS1 is ubiquitously active across normal somatic tissues of humans and other mammals (Moore et al, 2021; Cagan et al, 2022), and it has been shown to also contribute de novo mutations to the human germline (Moore et al, 2021; Rahbari et al, 2016). We thus reasoned that its increased hotspot propensity should be detectable across these normal tissues as it is across human tumours. To test this, we collected somatic mutations identified across human and mouse colonic crypts (Cagan et al, 2022), and de novo germline mutations identified across 7 datasets of family pedigrees (An et al, 2018; Halldorsson et al, 2019; C Yuen et al, 2017; Sasani et al, 2019; Goldmann et al, 2016; Rahbari et al, 2016). Employing the subsampling strategy described above (between 2500 and 30,000 mutations depending on the mutation rate and sample size in each dataset; Methods), we found a median of 18 observed hotspots of SBS1 across human colonic crypts (Fig. 5I), 50 across mouse colonic crypts (Fig. 5J), and 97 across de novo human germline mutations (Fig. 5K). Similarly, we found high hotspot propensity of SBS17a and b in Barrett's oesophagus-stomach pre-malignant lesions (Paulson et al, 2022) (Appendix Fig. S26). Thus, the mechanisms driving the clustering of mutations contributed by SBS17a/b and SBS1 along the genome are not restricted to cancer cells, but operate also in non-malignant tissues, including germ cells in the case of SBS1.

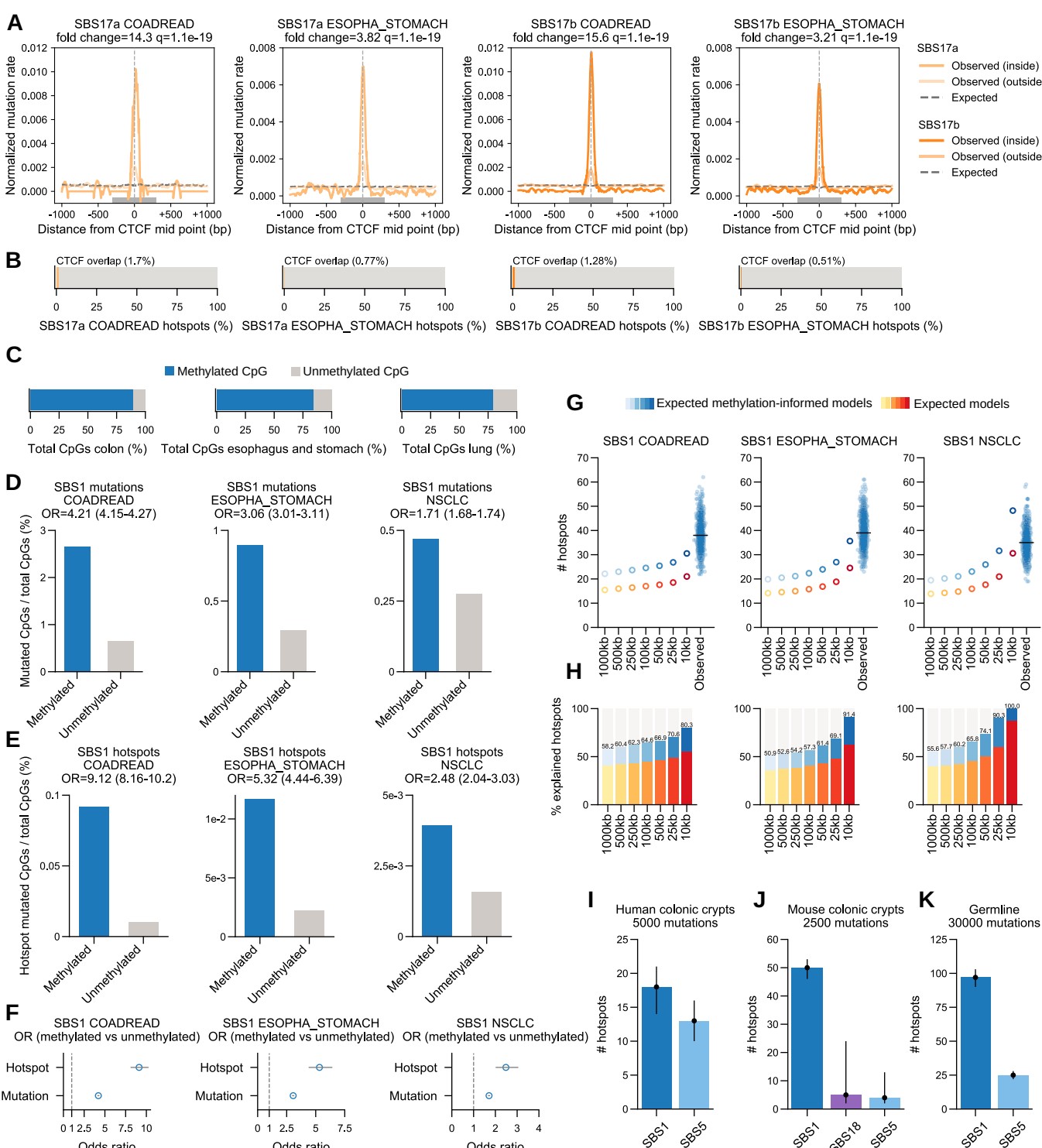

In summary, SBS17a/b and SBS1 are the signatures with the largest hotspot propensity. In the case of SBS17a and b, only 5–6% of their hotspot propensity can be explained by their mutational profile and the distribution of their mutations across 10 Kbp, suggesting that small-scale genomic features, including CTCF binding and others that remain to be identified, play a key role in their hotspot propensity. On the contrary, 80–100% of SBS1 observed hotspot propensity is driven by the relative scarcity and unequal distribution of methylated CpGs along the genome at 10 Kbp scale. Bearing in mind that the rate of hotspots is a readout of the variability of the mutation rate at base pair resolution, these findings are relevant for the correct modelling of SBS1 mutation rate to assess its contribution to the mutations in normal tissues of humans and mammals, and de novo mutations entering the germline.

**Figure 5. Underlying mechanisms of hotspot formation.**

(A) Piled-up normalised mutation rate of SBS17a and SBS17b across CTCF binding sites (600 bp; grey rectangle) and their flanking 5′ and 3′ regions (700 bp each). Mutations are split as overlapping hotspots (observed inside) and non-overlapping hotspots (observed outside). The fold change of inside mutation rates in CTCF binding sites versus flanks is shown on top. The significance of the fold change is computed by randomly distributing hotspot (inside) mutations across CTCF and flanking sequences considering the signature's trinucleotide mutational probability and the trinucleotide sequence composition (Methods). (B) Proportion of SBS17a and SBS17b hotspots in the cancer type overlapping CTCF binding sites. (C) Frequency of methylated and unmethylated CpGs in reference epigenomes (Methods; Dataset EV6). (D) Percentage of SBS1 mutated CpG sites among total methylated CpGs and total unmethylated CpGs in tissue-matched epigenomes for colorectal cancers ($n = 479$), oesophageal-stomach cancers ($n = 313$) and non-small cell lung cancers ($n = 413$). Odds ratio (OR) and its 95% confidence interval of mutations in methylated versus unmethylated CpGs are shown on top. (E) Percentage of CpGs with overlapping SBS1 hotspots in methylated and unmethylated CpGs, computed as in panel (D). Hotspots were retrieved from the cancer types shown in (D). (F) OR of SBS1 hotspots and mutations in methylated versus unmethylated CpG sites per cancer type as shown in (D) and (E). Error bars represent the 95% confidence interval of the OR. Sample sizes for the three cohorts are detailed in (D). (G) Comparison of observed versus expected SBS1 hotspot propensity. Expected data was generated using 300 mutations/sample and 100 samples. Orange dots depict the expected hotspot propensity using 1 Mbp to 10 Kbp models. Blue dots depict the expected hotspot propensity using methylation-informed 1 Mbp to 10 Kbp models. Observed hotspot propensity is shown as filled blue dots, where each dot represents a subsample ($n = 1000$) generated from 30,000 total mutations (300 mutations/sample, 100 samples). The median hotspot propensity is shown in black. Note that the expected hotspot propensity within 10 Kbp methylation-informed models is included in the range of the distribution of observed hotspot propensities for each cancer type. (H) Percentage of explained hotspots resulting from the expected models shown in (G) compared to the median observed hotspot propensity in the cancer type. Percentage of explained hotspots by methylation-informed models is shown above the bars. (I) Hotspot propensity in human colonic crypts computed using a total of 5000 mutations per signature (20 samples and 250 mutations/sample). Black dots show the median hotspot propensity across 1000 subsamples; error bars represent the IQR. (J) Hotspot propensity in mouse colonic crypts computed using a total of 2500 mutations per signature (25 samples and 100 mutations/sample). Black dots show the median hotspot propensity across 1000 subsamples; error bars represent the IQR. (K) Hotspot propensity in human de novo germline mutations computed using a total of 30,000 mutations per signature (30,000 random mutations from the merged datasets per subsample). Black dots show the median hotspot propensity across 1000 subsamples; error bars represent the IQR.

# Discussion

To study the variability of the mutation rate at single nucleotide resolution, here we introduced a new metric, hotspot propensity, which tracks the intrinsic tendency of mutational processes to form hotspots after correcting for differences in activity across cohorts and single samples. Measuring the hotspot propensity of 14 mutational signatures commonly active across tissues showed striking variability (up to 78 times fold) between them, with SBS1, SBS17a and SBS17b exhibiting the largest propensity to form mutational hotspots. The mutations of SBS1, the ubiquitous clock-like mutational process attributed to 5-methylcytosine deamination (Nik-Zainal et al, 2012; Alexandrov et al, 2013a), show a high propensity to form hotspots across most tissues analysed. SBS17a and b, of unknown aetiology, stand out as highly hotspot-prone both in primary oesophageal, stomach and colorectal tumours (Secrier et al, 2016). In our dataset, SBS17b hotspots are also observed in metastases of patients exposed to capecitabine/5-fluorouracil as part of the treatment of their primary tumours (Pich et al, 2019; Christensen et al, 2019) (Appendix Note 2). We corroborated the increased propensity to form hotspots—albeit at smaller rates than SBS1 and SBS17a/b—of the mutations of other signatures, including the UV-light caused SBS7a (Sabarinathan et al, 2016; Perera et al, 2016; Fredriksson et al, 2017; Elliott et al, 2018; Mao et al, 2018) and the APOBEC-related SBS2 (Buisson et al, 2019; Hess et al, 2019; Shi et al, 2020). We envisage that the increasing availability of cancer genome sequences, coupled with improvements in mutation calling and filtering of false positive calls, will pave the way for the exploration of hotspot propensity of other types of mutations not covered within our study (e.g., indels).

We have also leveraged the measurement of hotspot propensity to explore how much currently known determinants of mutagenesis actually contribute to the variability of the mutation rate at single nucleotide resolution. To do this, we built models of the expected hotspot propensity with increasing complexity—i.e., including additional known or suspected determinants of the mutation rate—across signatures and then compared them with the observed propensity. We were thus able to compute the accumulated contribution of these known or suspected large and small scale determinants of hotspot formation. The unevenness of the mutational profile of a signature, and the abundance of different trinucleotides in the human genome show a high contribution to the hotspot propensity of some mutational signatures. Notably, the high hotspot propensity of SBS1 is underpinned by its high activity at NpCpG trinucleotides, which are relatively depleted in the human genome (due to SBS1 action throughout evolution (Pich et al, 2018; Long et al, 2018)). Determinants of the mutation rate at relatively large scale—between 10 Kbp and 1 Mbp—show a differential contribution to hotspot propensity across signatures. For some signatures, like SBS2 and SBS1, large-scale seems to have a larger contribution than for others, such as SBS17a and SBS17b. Nevertheless, the sequence composition of the genome, the mutational profile of a signature, and the large-scale determinants of the mutation rate only explain a fraction, ranging from 5.2% in SBS17b to 68.3% in SBS1, of the observed hotspot propensity. This finding quantifies our knowledge gap of the determinants of the rate of mutations for different processes at nucleotide base resolution.

In order to help closing this gap, we explored the contribution of additional features to the high hotspot propensity of SBS17 and SBS1. We corroborated that the mutational hotspots of SBS17a/b are enriched for CTCF binding sites, although only a small fraction of these hotspots (0.5–1.7%) overlap CTCF binding sites. Thus indicating that other unknown factors, below the 10 Kbp scale, play a major role in the propensity of SBS17 mutations to form hotspots. For example, it has recently been suggested that nucleotide excision repair could play a role in SBS17 mutagenesis (Barbour et al, 2022). The situation presents in a totally different light for the hotspot propensity of SBS1 mutations. Careful calculation of the number of SBS1 expected hotspots accounting for its mutational profile, its mutation rate at 10 Kbp and the uneven distribution of methylation at CpG sites along the genome explain the vast majority (80–100%) of hotspots observed across three different tissues. In other words, our analysis allowed us to close the gap of explainability in the variability of the mutation rate of SBS1 at single nucleotide resolution. We foresee that the methodology described in the manuscript to compute the hotspot propensity of the mutations of different signatures will be applied to explore other potential new determinants of mutagenesis for signatures beyond SBS1, in particular, the elusive SBS17a/b.

While here we have focused on the study of the variability of the mutation rate at single nucleotide resolution in tumours, some of the mutational processes studied are also active in somatic healthy tissues and in the germline. A salient example is the spontaneous deamination of 5-methylcytosine, underlying SBS1 (Alexandrov et al, 2013a; Nik-Zainal et al, 2012), a universal age-related process affecting not only human somatic cells, but also human germ cells (Moore et al, 2021), and the tissues of other species (Cagan et al, 2022; Fryxell and Zuckerkandl, 2000). We show that, as is the case in colorectal tumours, a high propensity of hotspots may be observed for SBS1 mutations occurring in the normal colonic crypts of human and mouse. Likewise, SBS1 hotspots are found across de novo germline mutations. This suggests that the findings shown for SBS1 constitute a universal feature of somatic and germline tissues.

The driving motivation of this work was the systematic exploration of the variability of the mutation rate at single nucleotide resolution and its causes, which we undertook through the calculation of the propensity of the mutations of different processes to form hotspots. While the fine-grained variability of the mutation rate exploiting coding neutral hotspots has been previously explored across cancer genomes (Hess et al, 2019; Stobbe et al, 2019; Smith et al, 2016), to the best of our knowledge, this constitutes the first time hotspot propensity is exploited as a readout of mutation rate variability at nucleotide resolution. This high-resolution understanding of the mutation rate variability has different implications. First of all, it provides a new approach to study potential determinants of mutagenesis, as we have showcased with CpG methylation for SBS1. In the field of cancer genomics, statistical tools to identify signals of positive selection across the genome rely on the accuracy of background (i.e., neutral) mutagenesis. While state-of-the-art background models based on large and small scale features have been successful in the identification of protein-coding cancer drivers genes (Lawrence et al, 2013; Martincorena et al, 2017), this is still challenging for non-coding regions (Rheinbay et al, 2020; Sherman et al, 2022) and for individual cancer driver mutations (Muiños et al, 2021). The findings shown here could contribute to measuring the uncertainty of current background models at nucleotide resolution. Similarly, the analysis of evolutionary trajectories in cancer, normal tissues, and across evolution rely on the modelling of mutation rates at single nucleotide resolution. These analyses could also benefit from improving our estimates of the propensity of the different mutational processes to create hotspots as well as from studying new determinants of mutagenesis. We envision that our work constitutes a step forward in the pursuit of these long-term goals.

# Methods

## Reagents and tools

See Table 1.

**Table 1. Reagents and tools.**

| Reagent/resource | Reference or source | Identifier or catalogue number |
|---|---|---|
| **Datasets analysed** | | |
| Chromatin accessibility (DNase-seq) data from reference epigenomes | Roadmap Epigenomics Consortium et al, 2015 | https://egg2.wustl.edu/roadmap/web_portal (additional details in Dataset EV6) |
| CTCF ChIP-seq peaks from normal colon and oesophagus | Hammal et al, 2022 | https://remap2022.univ-amu.fr/ (additional details in Dataset EV6) |
| De novo human germline mutations | An et al, 2018; Halldorsson et al, 2019; Yuen et al, 2017; Sasani et al, 2019; Goldmann et al, 2016; Rahbari et al, 2016 | |
| Fractional methylation data (WGBS) from human reference epigenomes | Roadmap Epigenomics Consortium et al, 2015 | https://egg2.wustl.edu/roadmap/web_portal (additional details in Dataset EV6) |
| Gene expression (RNA-seq) from human reference epigenomes | Roadmap Epigenomics Consortium et al, 2015 | https://egg2.wustl.edu/roadmap/web_portal (additional details in Dataset EV6) |
| gnomAD hg38 population variants v3.0 | Karczewski et al, 2020 | https://gnomad.broadinstitute.org/ |
| Replication timing data (Repli-seq) from 7 human cell lines (Helas3, Hepg2, Huvec, Imr90, Mcf7, Nhek, and Sknsh) | ENCODE Project Consortium, 2012 | hgdownload.cse.ucsc.edu/goldenPath/hg19/encodeDCC/wgEncodeUwRepliSeq (additional details in Dataset EV6) |
| Somatic mutations from cancer cohorts | Pinto et al, 2015; Ma et al, 2015; ICGC/TCGA Pan-Cancer Analysis of Whole Genomes Consortium, 2020; Faber et al, 2016; Priestley et al, 2019; Lu et al, 2015; Parker et al, 2014; Tirode et al, 2014; Wu et al, 2014; Qaddoumi et al, 2016; Robinson et al, 2012; Hoang et al, 2018; Cheung et al, 2012; Pugh et al, 2013; Chen et al, 2014; Zhang et al, 2012; Chen et al, 2013; Gadd et al, 2017 | |
| Somatic mutations from normal colonic crypts from human and mouse | Cagan et al, 2022 | |
| Somatic mutations from Barrett's oesophagus | Paulson et al, 2022 | |

**Table 1.** (continued)

| Reagent/resource | Reference or source | Identifier or catalogue number |
|---|---|---|
| **Software** | | |
| bgreference v0.6 | https://bitbucket.org/bgframework/bgreference | |
| GEnomic Multi-tool (GEM) version 2013-04-06 | Marco-Sola et al, 2012<br>https://bio.tools/gemmapper | |
| HotspotFinder v1.0 | This study<br>https://bitbucket.org/bbglab/hotspotfinder | |
| Jupyter Notebook v5.0.0 | Kluyver et al, 2016<br>https://jupyter.org/ | |
| logomaker v0.8 | Tareen and Kinney, 2020<br>https://github.com/jbkinney/logomaker | |
| matplotlib v3.1 | Hunter, 2007<br>https://matplotlib.org/ | |
| numpy v.18 | Harris et al, 2020<br>https://numpy.org/ | |
| pandas v1.0 | Reback et al, 2022<br>https://pandas.pydata.org | |
| pybedtools v0.8 | Quinlan and Hall, 2010; Dale et al, 2011<br>https://daler.github.io/pybedtools | |
| pyBigWig v0.3.18 | Ryan et al, 2016<br>https://github.com/deeptools/pyBigWig | |
| pyliftover v0.3 | https://pypi.org/project/pyliftover | |
| Python v3.6 | https://www.python.org | |
| scipy v1.4 | Virtanen et al, 2020<br>https://scipy.org | |
| SigProfilerExtractor v1.1.0 | Islam et al, 2022<br>https://github.com/AlexandrovLab/SigProfilerExtractor | |
| SigProfilerMatrixGenerator v1.1.26 | Bergstrom et al, 2019<br>https://github.com/AlexandrovLab/SigProfilerMatrixGenerator | |
| statsmodels v0.11 | Seabold and Perktold, 2010<br>https://www.statsmodels.org | |
| **Other** | | |
| COSMIC Cancer Gene Census (24-08-2021) | Sondka et al, 2018 | https://cancer.sanger.ac.uk/census |
| COSMIC SBS GRCh38 Mutational Signatures v3.2 | Alexandrov et al, 2013b, 2020 | https://cancer.sanger.ac.uk/signatures/documents/453/COSMIC_v3.2_SBS_GRCh38.txt |
| ENCODE Unified GRCh38 Blacklist | ENCODE Project Consortium, 2012 | https://www.encodeproject.org/files/ENCFF356LFX |
| Gencode v35 | Frankish et al, 2019 | ftp.ebi.ac.uk/pub/databases/gencode/Gencode_human/release_35/gencode.v35.annotation.gtf.gz |
| IntOGen Compendium of Cancer Genes (01-02-2020 release) | Martínez-Jiménez et al, 2020 | https://www.intogen.org |
| OncoTree (02-11-2021 release) | Kundra et al, 2021 | https://oncotree.mskcc.org |

## Methods and protocols

### Somatic mutation data from cancer cohorts

We collected somatic mutations from 78 cohorts of whole genome sequenced cancer patients included in IntOGen (Martínez-Jiménez et al, 2020) (release 1 February 2020). Cohorts contained primary and metastatic tumours from adult and paediatric individuals, encompassing a total of 7507 samples and 83,410,018 somatic mutations (Pinto et al, 2015; Ma et al, 2015; ICGC/TCGA Pan-Cancer Analysis of Whole Genomes Consortium, 2020; Faber et al, 2016; Priestley et al, 2019; Lu et al, 2015; Parker et al, 2014; Tirode et al, 2014; Wu et al, 2014; Qaddoumi et al, 2016; Robinson et al, 2012; Hoang et al, 2018; Cheung et al, 2012; Pugh et al, 2013; Chen et al, 2014; Zhang et al, 2012; Chen et al, 2013; Gadd et al, 2017). Detailed information about each sequencing cohort and information on how to download them can be found at Dataset EV1 and www.intogen.org.

### Pre-processing of cancer cohorts

In order to homogenise the datasets for our analysis and minimise the number of false mutation calls, we pre-processed individual cohorts as follows:

Liftover of somatic mutations to GRCh38 reference genome: mutations in cohorts that used GRCh37 as reference genome were lifted over to GRCh38 using pyliftover package version 0.3 (pypi.org/project/pyliftover/) as described in (Martínez-Jiménez et al, 2020). Only mutations that mapped to GRCh38 were kept for analysis.

Filtering of somatic mutations: we removed mutations that (a) fell outside of autosomal or sexual chromosomes; (b) had the same reference and alternate nucleotides; (c) had a reference nucleotide that did not match the annotated hg38 reference nucleotide; (d) had an unknown nucleotide—a nucleotide not corresponding to A, C, G, T—in their trinucleotide or pentanucleotide reference sequence, as stated by their start position; (e) were classified as complex indels—indels that are a mixture of insertions and deletions such as $GTG > GAAA$.

Filtering of germline variants: our analysis aimed for the identification of hotspots of somatic mutations. In order to decrease contamination of somatic calls by unfiltered germline mutations, we removed mutations overlapping population variants. Briefly, we removed mutations overlapping genomic positions with one or more polymorphic variants (i.e., allele frequency equal or greater than 1%) (see 'Mappable genome and high mappability genomic bins' section for complete details).

Filtering of low mappability sequences: non-mappable regions (i.e., repetitive or non-unique sequences in the genome) are prone to sequencing artefacts. To control such errors, we discarded mutations that (1) fell outside high mappability regions and/or (2) overlapped blacklisted regions of low mappability (see 'Mappable genome and high mappability genomic bins' section for complete details).

Filtering of hypermutated samples: from each cohort, we filtered out hypermutated samples, this is, samples that carried more than 10,000 mutations and exceeded 1.5 times the interquartile range over the 75th percentile, as described in (Martínez-Jiménez et al, 2020).

### Cancer types classification

WGS samples were merged into 49 cancer types comprising one or more individual cohorts. Cancer type classification was based on the Memorial Sloan Kettering Cancer Centre (MSKCC) OncoTree (Kundra et al, 2021) (02-11-2021 release, available at onco-tree.mskcc.org). Assignment of each cohort to the different cancer type levels in the OncoTree hierarchy was carried out using the available clinical information of the cohort and can be found in Dataset EV1. Ad-hoc cancer types were added in those cases where the OncoTree classification did not fulfil the cohort definition. In order to avoid redundant cancer types—entities containing the same or very similar set of samples—within our analysis, we simplified the resulting hierarchy into two different levels A (specific) and B (meta-cancer type): level A entities were the most specific annotation available for a group of samples (e.g., melanomas); when two or more level A entities could be merged together according to the hierarchy, a level B annotation was added (e.g., melanomas, basal cell carcinomas, and cutaneous squamous cell carcinomas were grouped in skin cancers). Finally, all samples

were merged into the Pancancer level. Cancer types included in the analysis are listed in Dataset EV2.

### Pre-processing of cancer types

In order to identify the presence of multiple samples originating from the same donor, we conducted a systematic analysis of shared mutations among samples in each cohort and cancer type. Briefly, for every sample in a dataset, we computed the number of equal mutations with any other sample in the group and divided it over the total number of mutations of both samples in the comparison. Samples with more than 10% of shared mutations with any other sample were flagged for manual review. Two samples from M_OS were found to have primary tumour samples sequenced in D_OS and were subsequently removed from the metastatic cohort M_OS. The sample size of our analysis consisted of all available samples per cancer type after filtering.

### Driver gene annotations

Cancer driver genes were collected from two sources: the Compendium of Cancer Genes from the driver discovery pipeline IntOGen (Martínez-Jiménez et al, 2020) (release 01-02-2020) and the COSMIC Cancer Gene Census (Sondka et al, 2018) (CGC) (downloaded on 24-08-2021). The Compendium of Cancer Genes is composed of genes with experimental and/or in silico protein-coding driver evidence ($n = 568$ genes). CGC list contains expert-curated genes with experimental driver evidence from sporadic and familial cancers. Only those genes annotated as somatic and having a cancer role different from fusion partners were included ($n = 589$ genes). Our final set of driver genes consisted of 782 genes as listed in Dataset EV3. The genomic coordinates of the coding and surrounding non-coding sequences of driver genes were obtained from Gencode (Frankish et al, 2019) v35 comprehensive gene annotation file at ftp.ebi.ac.uk/pub/databases/gencode/Gencode_-human/release_35/gencode.v35.annotation.gtf.gz (downloaded on 30-08-2020) (Appendix Note 1).

### Mappable genome and high mappability genomic bins

We defined the mappable genome as the fraction of the reference hg38 genome included in our analysis (total nucleotides = 2,439,219,900 bp; nucleotides with known trinucleotide context = 2,439,219,170 bp). The mappable genome consisted of regions of high mappability that did not overlap with (1) blacklisted sequences of low mappability, (2) genomic positions containing population variants, and (3) coding or non-coding regions of driver genes (see 'Driver gene annotations' and Appendix Note 1 for further details). Regions of high mappability (≥0.9) based on 100-mer pileup mappability were computed for hg38 reference genome using The GEnomic Multi-tool (Marco-Sola et al, 2012) (GEM) mappability software version 2013-04-06. BED files containing hg38 blacklisted regions of low mappability were obtained from the ENCODE Unified GRCh38 Blacklist (downloaded from encodeproject.org/files/ENCFF356LFX on 16-06-2020) (ENCODE Project Consortium, 2012). Positions containing population variants were defined as those overlapping any substitution or short indel with total variant allele frequency above 1% as identified by gnomAD (Karczewski et al, 2020) version 3.0 (downloaded from gnomad.-broadinstitute.org on 25-06-2020). Next, we defined a set of high mappable megabases (1 Mbp bins) based on their overlap to the mappable genome. We first obtained hg38 genomic bins by

partitioning chromosome coordinates in consecutive non-overlapping chunks of 1 Mbp length. For each bin, we computed the sequence overlap with the mappable genome using the Python library pybedtools version 0.8 (Quinlan and Hall, 2010; Dale et al, 2011) and kept those autosomal bins whose sequence overlap with the mappable genome was above the first quartile of the distribution of fractional sequence overlap across megabase bins (Q1 = 0.80 fractional overlap; $n = 2196$ bins) (total nucleotides = 2,012,091,302 bp; nucleotides with known trinucleotide context = 2,012,091,115). For the set of mappable 1 Mbp bins, we obtained sets of sub-megabase bins of lengths 500 Kbp ($n = 4392$ bins), 250 Kbp ($n = 8784$ bins), 100 Kbp ($n = 21,960$ bins), 50 Kbp ($n = 43,920$ bins), 25 Kbp ($n = 87,840$ bins), and 10 Kbp ($n = 219,600$ bins) bins by partitioning individual 1 Mbp bins into consecutive chunks. All hg38 nucleotide sequences were retrieved from the Python package bgreference version 0.6.

### Identification of hotspots of somatic mutations

Genome-wide recurrently mutated positions from independent samples were identified using the new algorithm HotspotFinder version 1.0.0 (Appendix Note 1), freely available at bitbucket.org/bbglab/hotspotfinder. Hotspots of the four mutation types (SNVs, MNVs, insertions and deletions) were analysed separately. For each cancer type, HotspotFinder was run over the set of filtered mutations after excluding those overlapping coding or non-coding sequences (5′UTR, 3′UTR, splice sites, introns, proximal and distal promoters; Appendix Note 1) of driver elements. Hotspots were identified as single positions in the genome that contained (i) 2 or more mutations of equal alternates (e.g., two C>T transitions) or (ii) 2 or more mutations of different alternates (e.g., C>T and C>G). All the analyses included in the present work were carried out using hotspots of equal alternate. Hotspots were annotated with the default mappability, population variants and genomic regions provided within the method (Appendix Note 1) and those non-overlapping genomic elements were kept. All other parameters were set as default.

### Hotspot burden modelling

We modelled the relationship between hotspot burden and mutation burden per sample for each cancer type with univariate ordinary least squares (OLS) regression models using the Python package statsmodels version 0.11 (Seabold and Perktold, 2010).

### Estimation of conversion rates

Conversion rates or the number of mutations to observe 1 hotspot were calculated for cancer types with more than 100 individuals through a subsampling experiment. For 1000 times, we selected 100 random individuals (without replacement) and pooled their SNVs to identify hotspots of equal alternate as previously explained. We then modelled the number of hotspots per individual against their observed mutation burden using statsmodels OLS regression models (Seabold and Perktold, 2010). Conversion rates were computed as the inverse of the regression slope of significant models ($p < 0.05$) for those cancer types with at least 750 significant linear models across random replicates.

### Enrichment of substitution types in hotspots

Mutations overlapping and non-overlapping hotspots (mutations inside and outside hotspots, respectively) were classified into 6 and 96 pyrimidine-based substitution types based on GRCh38 reference genome using SigProfilerMatrixGenerator (Bergstrom et al, 2019) version 1.1.26. Hotspot enrichments for each substitution in a cancer type were computed as the ratio (fold change) of the substitution frequency in the set of mutations inside versus the substitution frequency outside. Clustering of cancer types according to enrichments of 6-class based substitutions was computed using the hierarchical clustering function cluster.hierarchy from the Python library scipy version 1.4 (Virtanen et al, 2020) with the linkage function 'complete'.

### Mutational signatures extraction

De novo trinucleotide-based SBS mutational signatures (96-mutation types using pyrimidines as reference) were extracted using SigProfiler framework (Bergstrom et al, 2019; Islam et al, 2022; Alexandrov et al, 2013b) for the 31 cancer types bearing at least 30 samples and 100,000 total SNVs (Appendix Note 2). Input GRCh38 96-mutational catalogues were calculated using SigProfilerMatrixGenerator (Bergstrom et al, 2019) version 1.1.26 and mutational signatures were extracted with SigProfilerExtractor (Islam et al, 2022) version 1.1.0 (Appendix Note 2). De novo signatures were decomposed into COSMIC v3.2 GRCh38 reference signatures to allow comparisons across cancer types (Appendix Note 2). All SBS signature names used in the manuscript correspond to this reference set. Signatures that were present in at least 5% of mutations in a sample were considered active in the sample. At the cancer type level, signatures that were active in at least 5% of samples were considered active in the cancer type.

### Assignment of mutational signatures to mutations and hotspots

The probability of each SNV—considering its sample of origin and trinucleotide context—to arise from each of the decomposed COSMIC signatures in the cancer type was obtained from SigProfilerExtractor ('Decomposed_Mutation_Probabilities.txt' table for the best extracted solution). As a result, a vector of mutational probabilities was generated for each SNV. In those cases where the 1 to 1 attribution of mutations to signatures was required, mutations were credited to the signature showing highest mutational probability (maximum likelihood (Morganella et al, 2016)). Hotspots were assigned to mutational signatures by computing, first, the average mutational probability vector among the mutations contributing to the hotspot, and then selecting the signature with the maximum average probability (check Appendix Note 3 for further details).

### Estimation of hotspot propensity

We set to estimate the propensity of commonly active mutational signatures to form hotspots across cancer types independently of their number of exposed samples and mutation burden contributed to each of them. First, we selected the 7 cancer types with the largest sample size (5000 or more observed hotspots and prioritising non-meta-cancer types when possible), including: bladder-urinary tract cancers (BLADDER_URI), breast cancers (BRCA), colorectal cancers (COADREAD), oesophagus-stomach cancers (ESOPHA_STOMACH), non-small cell lung cancers (NSCLC), prostate cancers (PROSTATE) and skin melanomas (SKCM). Then, we selected the 14 signatures that fulfilled the following criteria: (i) they showed 450 or more attributed hotspots (resulting in at least 1% of the total hotspots in the cancer type)

and (ii) contributed more than 300 high confidence mutations per sample attributed to the signature via highest probability (maximum likelihood $p > 0.5$) in at least 101 samples within the cancer type. The signature-cancer type pairs that passed these thresholds included: SBS1 (BLADDER_URI, BRCA, COADREAD, ESOPHA_STOMACH, NSCLC, PROSTATE), SBS13 (BLADDER_URI, BRCA, NSCLC), SBS17a (COADREAD, ESOPHA_STOMACH), SBS17b (COADREAD, ESOPHA_STOMACH), SBS18 (COADREAD), SBS2 (BLADDER_URI, BRCA, NSCLC), SBS3 (BRCA), SBS4 (NSCLC), SBS40 (COADREAD, NSCLC), SBS5 (BLADDER_URI, BRCA, COADREAD, ESOPHA_STOMACH, NSCLC, PROSTATE), SBS7a (SKCM), SBS7b (SKCM), SBS8 (PROSTATE), SBS93 (COADREAD). To estimate the propensity of a signature to form hotspots in a cancer type while accounting for differences in sample size and activity, we subsampled groups of 100 tumours of a given cancer type with a fixed number $n$ (between 100 and 300 mutations/sample or ~0.05 and 0.15 mutations/sample·Mbp), of randomly selected high confidence mutations (maximum likelihood $p > 0.5$) without replacement contributed by the signature under analysis. For each of the 1000 subsamples, we then counted the number of observed hotspots (allowing a single hotspot per position) among the $100 \cdot n$ subsampled mutations of the signature under analysis across cancer types. This analysis was carried out over the set of high mappable megabase bins ($n = 2196$ bins; 2,012,091,302 bp). To estimate hotspot propensity at a larger mutation rate, we subsampled 60,000 total mutations (100 samples and 600 mutations/sample or 0.3 mutations/sample·Mbp) for those signatures where at least 101 samples had 600 high confidence mutations in the cancer type. Due to the limited sample size, the following signature and cancer type pairs were not included in this analysis: SBS1-BLADDER_URI, SBS1-BRCA, SBS1-NSCLC, SBS5-BLADDER_URI, and SBS17a-COADREAD.

### Signatures enrichment in hotspots

For each sample containing hotspots, we first computed the frequency of its active signatures in the sets of hotspot and non-hotspot mutations. These frequencies were obtained by aggregating the signature mutational probability vectors (conveying the relative contribution of all possible signatures to a mutation) across all mutations in the specified set, which were subsequently normalised to 1 (see 'Assignment of mutational signatures to mutations and hotspots'). Then, a signature fold change (FC) in a given sample was computed as the ratio of the normalised frequency of the signature $S$ inside hotspots versus the normalised frequency of the signature outside hotspots:

$$\text{FC}(S) = \left( \frac{\text{Prob}(S)_{\text{inside}}}{\sum_T \text{Prob}(T)_{\text{inside}}} \right) / \left( \frac{\text{Prob}(S)_{\text{outside}}}{\sum_T \text{Prob}(T)_{\text{outside}}} \right)$$

To obtain the fold change per active signature we calculated the median fold change among active samples. For each signature, we tested whether the magnitude of the frequency inside hotspots deferred from that of non-hotspot mutations across sample-paired observations. We applied a two-sided Wilcoxon rank-sum test using the 'wilcoxon' function from the Python module scipy.stats (Virtanen et al, 2020). The obtained $p$-values were adjusted for multiple testing using the Benjamini–Hochberg method from statsmodels.sandbox.stats.multicomp function (Seabold and Perktold, 2010) in Python with $\alpha = 0.01$.

### Entropy of mutational signatures profiles

The entropy of the 96-channel profiles of SBS mutational signatures (COSMIC v3.2 GRCh38; available at cancer.sanger.ac.uk/signatures/documents/453/COSMIC_v3.2_SBS_GRCh38.txt) (Alexandrov et al, 2013b, 2020) was calculated after correcting by the trinucleotide content of the genome, i.e. we only account for the relative mutability of each context. This is done by taking the frequency profiles from COSMIC, dividing each trinucleotide frequency by the abundance of the corresponding reference triplet in the mappable genome plus the coding or non-coding regions of driver genes (which would yield the closest approximation to the full hg38 reference genome) and normalising the resulting profile so that the probabilities add up to 1 (Dataset EV5). The entropy, $H$, of a mutational signature profile given by a vector of frequencies ($p_k | k = 1, ..., 96$) was calculated using the scipy.stats.entropy function (Virtanen et al, 2020) in Python as follows:

$$H = -\sum_k p_k \cdot \log(p_k)$$

### Theoretical models of hotspots formation

We devised a method to compute the expected number of hotspots generated (i) by a given mutational process and (ii) in a given DNA region. The method assumes a simplifying theoretical scenario whereby (i) all samples in the cohort have identical mutation rates and (ii) positions mutate independently from one other. Briefly, if the regional mutation rate is homogeneous across the genome, we can estimate the probability that each position in the region undergoes each possible type of mutation (alternate allele) consistently with the relative mutability of each trinucleotide context dictated by the mutational process. The expected presence of a hotspot at a given position can then be calculated as the probability of at least two samples getting the same mutation at a given position. Because of statistical independence across positions, the regional expectation can be computed as the sum of expectations across positions.

Equipped with this method, we provide expected hotspot rate estimates associated with the 14 different mutational signatures from which we previously calculated hotspot propensity. Expected hotspot rates can be computed in two different scenarios: (i) genome wide: assuming that mutability is homogeneous along the genome, hence this is based on the trinucleotide composition of the genome alone, and (ii) per chunk: assuming cancer-type-specific variation in the distribution of mutation rate per signature asserted across genomic chunks. The analysis per chunks was conducted for different partitions of the mappable genome, using the following chunk sizes: 500, 250, 100, 50, 25, 10 Kbps and 1 Mbp (Appendix Note 5). The relative mutation rates per chunk, signature and cancer type were calculated from the total number of mutations attributed to the signature in the cancer type cohort estimated via maximum likelihood, i.e., mapping mutations to the signature with highest probability to have generated the mutation in the corresponding sample (Appendix Note 3).

The same signature-cancer type pairs used to calculate observed hotspot propensity were analysed. For both models, comparisons between observed and expected hotspot propensity were carried out using 30,000 total mutations (300 muts/sample across mappable 1 Mbp bins), which was equivalent to ~0.15 muts/Mbp. To match the

definition of hotspots with that of the expected calculation, only 1 observed hotspot per genomic position was considered (e.g., 2C>A and 2C>T mutations within the same position resulted in 1 hotspot).

We also implemented genome-wide and per-chunk theoretical hotspot rate models that used non-standard profiles to represent mutational processes. For the SBS1-induced hotspot analysis we implemented models that account for the differential SBS1 mutability across NpCpG>T contexts with and without accounting for the mutation rate bias due to methylated cytosines. For the models using methylated and unmethylated cytosine channels, methylated-unmethylated mutation rate fold-change was inferred empirically for each of the cancer types: COADREAD, ESO-PHA_STOMACH, NSCLC (see section 'Analysis of methylated CpG sites' in Methods and Appendix Note 5).

### Analysis of large-scale chromatin features

Chromatin accessibility and gene expression data for different tissues and cell lines matching the cancer types under analysis were obtained from the Epigenome Roadmap Project (Roadmap Epigenomics Consortium et al, 2015) at egg2.wustl.edu/roadmap/web_portal (see Dataset EV6 for complete details and URLs). Replication timing data for 7 cell lines from solid tissues was obtained from ENCODE (ENCODE Project Consortium, 2012) at hgdownload.cse.ucsc.edu/goldenPath/hg19/encodeDCC/wgEncodeUwRepliSeq. For each of these three chromatin features (mapped to hg19 reference genome), we obtained the average signal across hg19 mappable megabases—liftovered from the hg38 mappable megabase coordinates—as follows. For chromatin accessibility, we first computed the average counts across megabases from genome-wide fold-enrichment DNase counts tracks (BIGWIG format) per epigenome. Then, for each megabase, we computed the average DNase-seq signal across the different epigenomes linked to a cancer type. Similarly, megabase gene expression signals were computed from normalised coverage genome tracks (BIGWIG format). In this case, if stranded libraries were available for an epigenome, we first added up the absolute RNA-seq signals from the negative and positive strands and then computed the average signal per epigenome and megabase. The cancer type signal per megabase was obtained by calculating the mean megabase RNA-seq signal from the different epigenomes linked to the cancer type. For replication timing data, we used the percentage-normalised Repli-seq signal tracks (BIGWIG format). Following the same approach, we obtained the average Repli-seq signal across cell lines per megabase. Signals extracted from BIGWIG files were handled using the Python package pyBigWig version 0.3.18 (Ryan et al, 2016).

In order to investigate the relationship between the number of hotspots and non-hotspot mutations with chromatin accessibility, gene expression and replication timing per signature and cancer type, we first intersected mutations inside and outside hotspots with mappable megabase bins. Next, we added up the vector of mutational probabilities of each mutation to arise from a signature in the cancer type (see 'Assignment of mutational signatures to mutations and hotspots'). For each signature, we normalised its signal across megabases for mutations inside and outside hotspots. We then categorised mappable megabases into 10 percentiles (deciles) according to the distribution of the chromatin feature signal across megabases and plotted the signature activity inside and outside hotspots on each decile.

### Overdispersion of mutational signatures across the genome

For each signature, we modelled the distribution of their attributed mutation counts across mappable 1 Mbp and 500–10 Kbp bins by fitting a negative binomial regression model, which yields an overdispersion parameter. Briefly, the overdispersion parameter, referred to as $\alpha$ throughout, measures the excess variance over the mean, i.e., excess variance over the variance that we would expect if the mutation counts were Poisson distributed. Specifically, if $\mu$ denotes the mean, in our negative binomial regression setting the relationship between the variance $v$ and the mean $\mu$ is given by the equation:

$$v = \mu + \alpha\mu^2$$

The Python function statsmodels.discrete.discrete_model.NegativeBinomial was used (Seabold and Perktold, 2010).

### Analysis of CTCF binding sites

CTCF binding sites were defined as CTCF ChIP peaks in a tissue or cell line matching the cancer type. hg38 ChIP peak coordinates were downloaded from ReMap2022 (Hammal et al, 2022) at remap.univ-amu.fr on 08-03-2022 (Dataset EV6). Only CTCF peaks within autosomes, 200–600 bp long, and showing 90% or more overlap to the mappable genome were kept for analysis (sigmoid colon $n = 27,626$; epithelial oesophagus $n = 38,740$). Intersections between annotations were carried out using pybedtools (Quinlan and Hall, 2010; Dale et al, 2011). The enrichment of hotspots within CTCF binding sites was computed as follows. For each individual CTCF feature, we first constructed a window of length $L$ where the feature of length $L_{feature}$ was positioned in the centre surrounded by two flanking sites of equal size $(L-L_{feature})/2$. Window length $L$ was defined as 2000 bp and feature length $L_{feature}$ as 600 bp (the maximum size encompassing all individual features under analysis). For each cancer type, we intersected each CTCF window with inside and outside hotspot mutations attributed to SBS17a and b by maximum likelihood. We obtained the expected distribution of mutations per signature by randomising 1000 times the observed number of mutations inside hotspots across the window according to the signature trinucleotide probabilities and the window sequence composition. To compute the hotspot enrichment in CTCF binding sites (fold change), we piled-up mutations in equal positions across windows and calculated the ratio of mutations inside the feature versus its flanks. The significance of the observed fold changes was estimated by fitting simulated fold changes to a gaussian kernel density estimate distribution and deriving the upper quantile of the observed fold change. Resulting $p$-values were adjusted for multiple testing using the statsmodels Benjamini–Hochberg function (Seabold and Perktold, 2010) with $\alpha = 0.01$. To visualise the results across piled-up CTCF binding sites, observed and expected mutation counts per position were normalised to the respective total number of mutations in the set across the window. A smoothing Savitzky-Golay filter of length 101 bp was applied using the scipy.signal.savgol_filter function (Virtanen et al, 2020).

### Analysis of methylated CpG sites

Fractional methylation from whole genome bisulfite sequencing (WGBS) data for colon, oesophagus-stomach, and lung reference epigenomes was downloaded from Roadmap (Roadmap Epigenomics Consortium et al, 2015) on 12-11-2022 (Dataset EV6). These tissues were selected based on the availability of uniformly processed WGBS, the gold-standard assay to quantify genome-wide methylation, within the Roadmap project. Original methylation calls per CpG site were

liftovered from hg19 to hg38. Those failing to liftover or falling in a non-CpG site were removed. When different hg19 CpG entries mapped to an equal hg38 CpG site, only the first hg19 CpG entry was kept. The analysis was carried out using only CpGs within autosomes and overlapping mappable 1 Mbp bins. CpG sites were intersected with mutations and hotspots attributed to SBS1 in colorectal, oesophageal-stomach, and non-small cell lung cancers, and were subsequently annotated with the average fractional methylation across the epigenomes matching the tissue of origin of each cancer type. Those CpG sites with missing methylation data in one of the tissue-matched epigenomes were skipped from the analysis. CpGs with average fractional methylation above 0.5 were classified as methylated.

### Estimation of hotspot propensity in normal and germline tissues

We obtained somatic single base substitutions from different datasets to estimate SBS1 and SBS17a and b hotspot propensity in non-cancerous tissues. Somatic mutations from normal colonic crypts from human and mouse (Cagan et al, 2022) were downloaded from github.com/baezortega/CrossSpecies2021 and preprocessed as stated by the authors (steps 0 and 1 from the code repository). Only samples passing the quality criteria were used, resulting in 28 and 43 human and mouse samples, respectively. Somatic mutations from Barrett's oesophagus-stomach (BE) were obtained from (Paulson et al, 2022). This dataset contains samples from 80 BE patients collected at different time points. In order to analyse a single sample per patient, we randomly selected 1 sample from the first sampling time point (T1) in each patient, resulting in 80 BE samples. Germline de novo mutations were obtained from 7 different datasets (An et al, 2018; Halldorsson et al, 2019; C Yuen et al, 2017; Sasani et al, 2019; Goldmann et al, 2016; Rahbari et al, 2016), comprising a total of 7796 families. Of note, de novo mutations in these datasets were not mapped to individuals/samples of origin. All datasets containing human mutations mapped to hg19 were liftovered to the hg38 reference genome. Mutations failing to map to hg38 or being mapped to a different reference nucleotide in hg38 were discarded. Mutations in sexual chromosomes were removed in all datasets. Finally, we filtered all human datasets with our mappable genome 1 Mbp bins. After filtering, we obtained a total of 52,233, 37,691 and 911,922 somatic mutations from human colonic crypts, mouse colonic crypts, and BE samples, respectively. Filtering of de novo germline mutations resulted in a total of 581,978 mutations for analysis.

We identified the signatures active in human and mouse colonic crypts, as well as in the germline, by signature fitting. Unlike the de novo signature deconstruction methods where the mutational signatures are inferred from scratch, signature fitting assumes that a collection of mutational signatures is given to then infer an exposure per signature and sample. We implemented signature fitting as a non-negative least squares regression, whereby a linear model is fit with the least squares criterion and the estimated model parameters (exposures per signature per sample) are constrained to be non-negative. Concretely, if $S$ is a $96 \times k$ matrix with the $k$ given signature profiles arranged as columns, and $M$ is a $96 \times n$ matrix of mutation counts from $n$ samples, signature fitting solves for:

$$\hat{E} = argmin_{E \geq 0} |SE - M|_F^2$$

For human and mouse colonic crypts, we based our fitting on the three mutational signatures most active across the original study (Cagan et al, 2022), SBS1, SBS5 and SBS18. COSMIC v3.2

hg38 and mm10 reference signatures were used in human and mouse, respectively. In the case of the germline datasets, we tested the 92,170 possible combinations of 1–4 signatures from selected COSMIC signatures (SBS1, SBS2, SBS3, SBS4, SBS5, SBS6, SBS7a, SBS7b, SBS7c, SBS7d, SBS8, SBS9, SBS10a, SBS10b, SBS10c, SBS10d, SBS11, SBS12, SBS13, SBS14, SBS15, SBS16, SBS17a, SBS17b, SBS18, SBS19, SBS20, SBS21, SBS22, SBS23, SBS24, SBS25, SBS26, SBS27, SBS28, SBS29, SBS30, SBS40, SBS44). We conducted signature fitting with each combination and we thereupon computed the Akaike Information Criterion (AIC) of the resulting models. We selected the solution containing the signatures that were most frequently found across the top best solutions (SBS1 and SBS5). In Barrett's oesophagus-stomach samples, we used the mutational probabilities of individual mutations to arise from the different signatures active in the dataset as provided by the authors (Paulson et al, 2022). We focused our analysis on the top 5 active signatures: SBS1, SBS5, SBS17b, SBS17a, and SBS18.

Finally, we estimated hotspot propensity in each dataset by randomly sampling a number of mutations and/or samples for 1000 times. A total of 5000 mutations per signature (20 samples and 250 mutations/sample) and 2500 mutations per signature (25 samples and 100 mutations/sample) were used in the human and mouse colonic crypts, respectively. For Barrett's oesophagus-stomach samples, we leveraged 15,000 mutations per signature (50 samples and 300 mutations/sample). To calculate germline hotspot propensity, where de novo mutations could not be mapped to individuals/samples of origin, we merged de novo mutations in the 7 datasets and randomly sampled 30,000 mutations.

### Sequence logos

We computed the frequency of the nucleotide sequence composition around hotspots attributed to a signature across cancer types. The same signature-cancer type groups as those used to calculate hotspot propensity were used. For each hotspot in the signature, we retrieved the 10 bp 5′ and 3′ flanking sequences (considering the strand containing a pyrimidine in the hotspot position) from bgreference v0.6 and built a 21 bp window centred at the hotspot. We then computed the information content over the nucleotide frequency with respect to the nucleotide hg38 mappable genome frequency at each position across the window. Logo plots and information content were generated using the Python package logomaker version 0.8 (Tareen and Kinney, 2020).

### Additional software used

The following packages were used across different analyses, including: Jupyter notebook version 5.0.0 (Kluyver et al, 2016), matplotlib version 3.1 (Hunter, 2007), numpy version 1.18 (Harris et al, 2020), and pandas version 1.0 (Reback et al, 2022).

## Data availability

The datasets analysed within this manuscript have been retrieved from elsewhere and are referenced in the manuscript, Extended View Datasets, Reagents and tools table, and Methods and protocols. The datasets and computer code produced in this study are available in the following databases: Hotspots identified across the cancer types analysed in the manuscript have been deposited at a *Zenodo* repository: https://doi.org/10.5281/zenodo.10004773 (https://doi.org/

10.5281/zenodo.10004773); HotspotFinder algorithm developed within this study is available at a Bitbucket repository: bitbucket.org/bbglab/hotspotfinder (https://bitbucket.org/bbglab/hotspotfinder); The in-house source code and data to reproduce the figures is available at a Github repository: github.com/bbglab/hotspot_propensity (https://github.com/bbglab/hotspot_propensity).

## Peer review information

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

## Acknowledgements

The authors wish to acknowledge the contribution of patients, families, and biomedical researchers who shared, processed, and sequenced the data used within the study. The results published here are in part based on the data generated by the Pan-Cancer Analysis of Whole Genomes, St. Jude Children's Research Hospital, PedcBioPortal, Therapeutically Applicable Research to Generate Effective Treatments programme. This publication and the underlying study have been made possible partly on the basis of the data that Hartwig Medical Foundation has made available to the study. We acknowledge the technical contributions of Iker Reyes-Salazar and Loris Mularoni to HotspotFinder algorithm and annotations. NL-B acknowledges funding from the European Research Council (consolidator grant 682398) and the European Regional Development Fund/Spanish Ministry of Science, Innovation and Universities —Spanish State Research Agency/DamReMap Project (RTI2018-094095-B-I00) and Fundación Científica Asociación Española Contra el Cáncer (AECC) (GC16173697BIGA). This work has also been supported by the project "Discovering the molecular signatures of cancer PROMotion to INform prevENTion" (PROMINENT) funded by Cancer Research UK (CGCATF-2021/100008), National Cancer Institute (1OT2CA278668-01) and the Spanish Cancer Association, AECC. IRB Barcelona is a recipient of a Severo Ochoa Centre of Excellence Award from the Spanish Ministry of Economy and Competitiveness (MINECO; Government of Spain) and is supported by CERCA (Generalitat de Catalunya). CA-P was supported by "la Caixa" Foundation (ID 100010434) fellowship (LCF/BQ/ES18/11670011).

## Author contributions

**Claudia Arnedo-Pac**: Conceptualisation; Data curation; Software; Formal analysis; Validation; Visualisation; Methodology; Writing—original draft; Writing—review and editing. **Ferran Muiños**: Software; Formal analysis; Validation; Visualisation; Methodology; Writing—original draft; Writing—review and editing. **Abel Gonzalez-Perez**: Conceptualisation; Supervision; Writing—original draft; Writing—review and editing. **Nuria Lopez-Bigas**: Conceptualisation; Resources; Supervision; Funding acquisition; Writing—original draft; Project administration; Writing—review and editing.

## Disclosure and competing interests statement

The authors declare no competing interests.

# Expanded View Figures

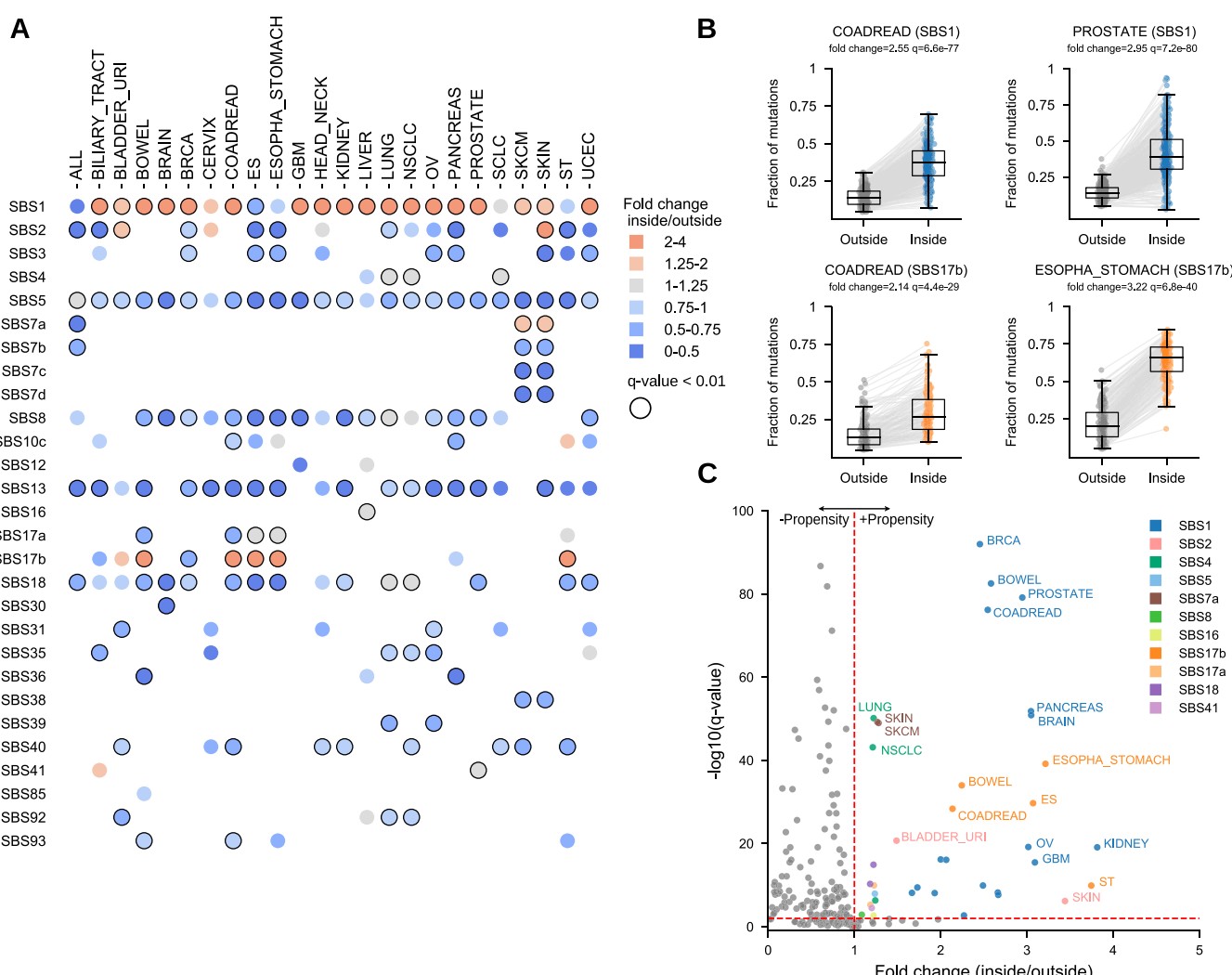

**Figure EV1.  Propensity of signatures to form hotspots across cancer types measured through inside-to-outside fold-change of their activity.**

(**A**) Heatmap showing the fold-change of mutational frequencies inside hotspots versus outside hotspots for each active signature in a cancer type. Signatures enriched in hotspots show fold-changes greater than 1 (grey) or 1.25 (red); signatures that are depleted in hotspots show fold-changes smaller than 1 (blue). The differences in signature frequency between inside and outside mutations were compared using two-sided Wilcoxon rank-sum test followed by Benjamini–Hochberg multiple testing correction (Methods). Significant fold-changes after multiple testing correction (*q*-value < 0.01) are shown in bold. (**B**) Illustration of Wilcoxon rank-sum test of inside-to-outside signatures activity. Boxplots depicting signature frequencies per sample across mutations outside and inside hotspots. Boxplot centre depicts the median, and the lower and upper bounds of the box represent the 1st and the 3rd quartiles, respectively. Whiskers extend 1.5 times the IQR below and above 1st and 3rd quartiles of the distribution. Dots depict samples where the signature is active (*n* = 465 for SBS1 and colorectal cancers; *n* = 486 for SBS1 and prostate cancers; *n* = 172 for SBS17b and colorectal cancers; *n* = 237 for SBS17b and oesophageal-stomach cancers; Methods). Grey lines connect data originating from the same sample. Fold changes and *q*-values computed from these comparisons are shown in panels (**A**) and (**C**). (**C**) Scatter plot showing the propensity of signatures to form hotspots computed as the fold-change and significance between their activities outside and inside hotspots per cancer type. Signature-cancer type pairs showing significant differences (*q*-value < 0.01) and fold change greater than 1 are shown in colour.

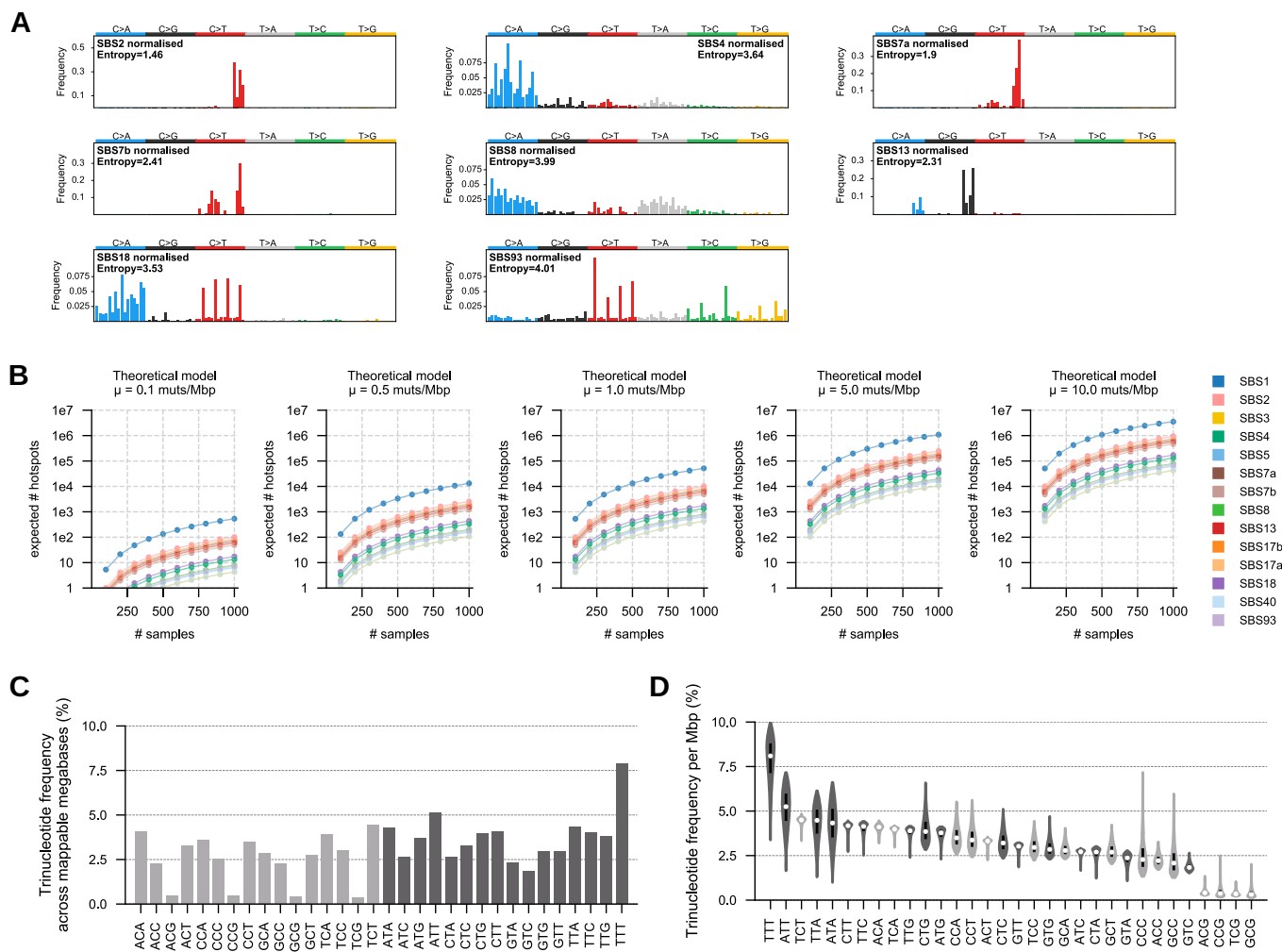

**Figure EV2. Influence of trinucleotide frequencies on hotspot propensity.**

(A) Normalised trinucleotide profiles and entropies from the 8 additional selected signatures under analysis (see Methods). (B) Theoretical number of expected hotspots across different mutation rates (0.1–10 mutations per sample per megabase) and sample sizes (100–1000 samples) for each of the 14 signatures. Theoretical estimates were calculated using the model of homogeneous distribution of trinucleotide-specific mutation rates across the genome (Methods; Appendix Note 5). Only positions within the mappable megabases were considered. The chosen mutation rates reflect a wide range of observed mutation rates across malignancies (Lawrence et al, 2013; Alexandrov et al, 2013a). (C) Bar plot showing the frequency of 32 pyrimidine-based trinucleotides across mappable genome megabases (n = 2196). (D) Violin plots showing the distribution of trinucleotides frequency across mappable megabases (n = 2196). White dots show the median trinucleotide frequency among megabases. Vertical black lines depict the 1st and 3rd quartiles of the distribution.

