## [Peer Review File · Molecular Systems Biology]

Hotspot propensity across mutational processes

Claudia Arnedo-Pac, Ferran Muinos, Abel Gonzalez-Perez, and Nuria Lopez-Bigas

DOI: 10.15252/msb.202312045

Corresponding author(s): Nuria Lopez-Bigas (nuria.lopez@irbbarcelona.org) , Abel Gonzalez-Perez (abel.gonzalez@irbbarcelona.org)

Review Timeline:

Submission Date:	5th Oct 23
Editorial Decision:	12th Oct 23
Revision Received:	30th Oct 23
Accepted:	9th Nov 23

Editor: Maria Polychronidou

Transaction Report:

The reviewers' comments and authors' responses are not available with this article, as the initial review process took place with another journal.

12th Oct 2023

Manuscript Number: MSB-2023-12045

Title: Hotspot propensity across mutational processes

Dear Nuria,

Thank you again for submitting your work to Molecular Systems Biology. I have now read your manuscript, the referee reports from the other two journals and your responses to these comments and discussed them with the team. Overall, we think that the review process has been thorough and therefore we can editorially assess the study using these previous reports, and without reviewing it from scratch. We think that all technical and other core concerns seem to have been addressed in the previous rounds of revision at the other two journals. Regarding the remaining concerns of reviewer #2, we think that your responses sound reasonable and therefore these concerns do not preclude publication in Molecular Systems Biology. As such, we have decided to proceed with publishing the study in Molecular Systems Biology without requesting further revisions.

We would only ask you to fix some editorial issues listed below, so that we can formally accept the study for publication.

- Please provide 5 keywords.
- Please include a Disclosure and Competing Interests Statement in the main text.
- Please provide a .doc version of the manuscript text (including legends for main figures and tables) and individual files for the main figures. The figure legends should be included at the end of the manuscript text, after the References.
- The Reference list should be formatted according to the Molecular Systems Biology style i.e. ordered alphabetically and listing the first 10 authors followed by et al.
- We have replaced Supplementary Information by the Expanded View (EV format). In this case, all additional figures can be included in a PDF called Appendix. Appendix figures should be labeled and called out as: "Appendix Figure S1, Appendix Figure S2... Appendix Table S1..." etc. Each legend should be below the corresponding Figure/Table in the Appendix. Please include a Table of Contents in the beginning of the Appendix. For detailed instructions regarding expanded view please refer to our Author Guidelines: .
- Supplementary Notes should be included in the Appendix and renamed to Appendix Notes.
- Supplementary Tables S1-S7 should be provided as EV Datasets (either as .xls files or .zip folders). Please provide one file per EV Dataset. Please include the description of each EV Dataset in the dataset file itself, ie. in a separate tab for .xls files or as a README.txt file in .zip folders.
- Please provide a "standfirst text" summarizing the study in one or two sentences (approximately 250 characters), three to four "bullet points" highlighting the main findings and a "synopsis image" (550px width and max 400px height, jpeg format) to highlight the paper on our homepage.
- All Materials and Methods need to be described in the main text. We would encourage you to use 'Structured Methods', our new Materials and Methods format. According to this format, the Material and Methods section should include a Reagents and Tools Table (listing key reagents, experimental models, software and relevant equipment and including their sources and relevant identifiers) followed by a Methods and Protocols section in which we encourage the authors to describe their methods using a step-by-step protocol format with bullet points, to facilitate the adoption of the methodologies across labs. More information on how to adhere to this format as well as downloadable templates (.doc or .xls) for the Reagents and Tools Table can be found in our author guidelines: . An example of a Method paper with Structured Methods can be found here:
- Please include a Data availability section describing how the data, code etc. generated in this study have been made available. This section needs to be formatted according to the example below:
The datasets and computer code produced in this study are available in the following databases:
- Chip-Seq data: Gene Expression Omnibus GSE46748 (<https://www.ncbi.nlm.nih.gov/geo/query/acc.cgi?acc=GSE46748>)
- [data type]: [full name of the resource] [accession number/identifier] ([doi or URL or identifiers.org/DATABASE:ACCESSION])
- The Data Availability section should be reserved only for data generated in the study. Data retrieved from other sources (e.g. previously published data) should be referenced/described in a separate section titled "Datasets analysed".
- Molecular Systems Biology supports formal data citations in the Reference list, to cite previously published datasets. In addition to citing the original papers that reported the data, we encourage you to also cite the relevant datasets directly in the Reference

list. In the text, references to datasets are included as "Data ref: Smith et al, 2001" or "Data ref: NCBI Sequence Read Archive PRJNA342805, 2017". In the Reference list, data citations are very similar to normal literature references but must be labeled with "[DATASET]" at the end of the reference. For detailed instructions please refer to our Author Guidelines .

- When you resubmit your manuscript, please download our CHECKLIST (<https://bit.ly/EMBOPressAuthorChecklist>) and include the completed form in your submission. *Please note* that the Author Checklist will be published alongside the paper as part of the transparent process (<https://www.embopress.org/page/journal/17444292/authorguide#transparentprocess>)

Please resubmit your revised manuscript online, with a covering letter listing amendments and responses to each point raised by the referees. Please resubmit the paper ****within one month**** and ideally as soon as possible. If we do not receive the revised manuscript within this time period, the file might be closed and any subsequent resubmission would be treated as a new manuscript. Please use the Manuscript Number (above) in all correspondence.

Click on the link below to submit your revised paper.

Kind regards,

Maria

Maria Polychronidou, PhD
Senior Editor
Molecular Systems Biology

If you do choose to resubmit, please click on the link below to submit the revision online before 11th Nov 2023.

IMPORTANT: When you send your revision, we will require the following items:

See also figure legend guidelines: <https://www.embopress.org/page/journal/17444292/authorguide#figureformat>

- Please note that corresponding authors are required to supply an ORCID ID for their name upon submission of a revised manuscript (EMBO Press signed a joint statement to encourage ORCID adoption).

(<https://www.embopress.org/page/journal/17444292/authorguide#editorialprocess>)

Currently, our records indicate that the ORCID for your account is 0000-0003-4925-8988.

Link Not Available

The system will prompt you to fill in your funding and payment information. This will allow Wiley to send you a quote for the article processing charge (APC) in case of acceptance. This quote takes into account any reduction or fee waivers that you may be eligible for. Authors do not need to pay any fees before their manuscript is accepted and transferred to the publisher.

***** PLEASE NOTE ***** As part of the EMBO Press transparent editorial process initiative (see our Editorial at <https://dx.doi.org/10.1038/msb.2010.72> , Molecular Systems Biology will publish online a Review Process File to accompany accepted manuscripts. When preparing your letter of response, please be aware that in the event of acceptance, your cover letter/point-by-point document will be included as part of this File, which will be available to the scientific community. More information about this initiative is available in our Instructions to Authors. If you have any questions about this initiative, please contact the editorial office (msb@embo.org).

All editorial and formatting issues were resolved by the authors.

9th Nov 2023

Manuscript number: MSB-2023-12045R

Title: Hotspot propensity across mutational processes

Dear Nuria,

Thank you again for sending us your revised manuscript. We are now satisfied with the modifications made and I am pleased to inform you that your paper has been accepted for publication.

Kind regards,

Maria

Maria Polychronidou, PhD
Senior Editor
Molecular Systems Biology
